# Stereotyping in the digital age: Male language is "ingenious", female language is "beautiful" – and popular

Tabea Meier[1,2]*, Ryan L. Boyd[3], Matthias R. Mehl[4], Anne Milek[5], James W. Pennebaker[6], Mike Martin[1,2,7], Markus Wolf[1], Andrea B. Horn[1,2]*

1 Department of Psychology, University of Zurich, Zurich, Switzerland, 2 University Research Priority Program (URPP) "Dynamics of Healthy Aging", University of Zurich, Zurich, Switzerland, 3 Department of Psychology, Lancaster University, Lancaster, Lancashire, United Kingdom, 4 Department of Psychology, University of Arizona, Tucson, Arizona, United States of America, 5 Department of Psychology, University of Münster, Münster, North Rhine-Westphalia, Germany, 6 Department of Psychology, The University of Texas, Austin, Texas, United States of America, 7 Collegium Helveticum, ETH Zurich, Zurich, Switzerland

* t.meier@psychologie.uzh.ch (TM); a.horn@psychologie.uzh.ch (ABH)

## Abstract

The huge power for social influence of digital media may come with the risk of intensifying common societal biases, such as gender and age stereotypes. Speaker's gender and age also behaviorally manifest in language use, and language may be a powerful tool to shape impact. The present study took the example of TED, a highly successful knowledge dissemination platform, to study online influence. Our goal was to investigate how gender- and age-linked language styles–beyond chronological age and identified gender–link to talk impact and whether this reflects gender and age stereotypes. In a pre-registered study, we collected transcripts of TED Talks along with their impact measures, i.e., views and ratios of positive and negative talk ratings, from the TED website. We scored TED Speakers' ($N$ = 1,095) language with gender- and age-morphed language metrics to obtain measures of female versus male, and younger versus more senior language styles. Contrary to our expectations and to the literature on gender stereotypes, more female language was linked to higher impact in terms of quantity, i.e., more talk views, and this was particularly the case among talks with a lot of views. Regarding quality of impact, language signatures of gender and age predicted different types of positive and negative ratings above and beyond main effects of speaker's gender and age. The differences in ratings seem to reflect common stereotype contents of warmth (e.g., "beautiful" for female, "courageous" for female and senior language) versus competence (e.g., "ingenious", "informative" for male language). The results shed light on how verbal behavior may contribute to stereotypical evaluations. They also illuminate how, within new digital social contexts, female language might be uniquely rewarded and, thereby, an underappreciated but highly effective tool for social influence. *WC = 286 (max. 300 words).*

**Data Availability Statement:** Our data and data analysis scripts are available on the Open Science Framework: https://osf.io/qkm6u/

**Funding:** Financial support by the Jacobs Foundation helped to conduct this research (https://jacobsfoundation.org/en/; doctoral fellowship awarded to TM) ABH received financial support by the Swiss National Science Foundation (https://www.snf.ch/en/; 1 grant: SNF PMPDP1_164470). Preparation of this manuscript was additionally aided by grants from the National Institutes of Health (https://www.nih.gov; 5R01GM112697-02 awarded to JWP, RLB), John Templeton Foundation (https://www.templeton.org; 2 grants #48503 and #61156 awarded to JWP, RLB), the Federal Bureau of Investigation (https://www.fbi.gov; 1 grant 15F06718R0006603 awarded to JWP, RLB), and the National Science Foundation (https://www.nsf.gov; 1 grant IIS-1344257 awarded to JWP, RLB). The views, opinions, and findings contained in this document are those of the authors and should not be construed as position, policy, or decision of the aforementioned agencies, unless so designated by other documents. The funders had no role in study design, data collection and analysis, decision to publish, or preparation of the manuscript.

**Competing interests:** The authors have declared that no competing interests exist.

# Introduction

A large part of social interaction nowadays takes place online and the digital age has brought new opportunities to interact and communicate with increasingly large audiences. Digital communication platforms represent modern contexts in which social processes naturally unfold, and they provide rich sources to study basic human behaviors, such as communication and social evaluation [1–3]. One of the main goals behind digital social media platforms is social influence. In fact, the influence that can be achieved through social media is at an unprecedented scale–content shared online can essentially reach billions of users.

It seems plausible that basic psychological processes, such as social influence and evaluation operate in comparable ways in online and offline settings. Evaluations in offline interactions are often biased in terms of social stereotyping. For example, women and older people tend to be disadvantaged in evaluations of expertise and authority [4]. The power of influence seen in digital communication domains may then come with both risks and opportunities to either reinforce or attenuate effects of such biases. In line with an empirically well-supported pro-male bias in evaluations [e.g., 5,6], past research indeed indicated that female speakers' talks shared on video platforms were less influential than male speakers' talks [7].

Salient social categories such as a person's gender and age, however, are but one of many bases for social processes. Behaviors commonly shown by social groups may additionally guide stereotypical beliefs and evaluations [8,9]. One behavioral manifestation that plays a major role in social evaluations is language use [e.g; 10]. As an example, texts written in a typical female style were evaluated as less competent compared to those in male style [11]. Findings like these support the assumption that language patterns are one behavioral feature that makes social groups such as gender or age salient and trigger stereotyping. Gender- and age-linked language have been quantified by deriving general language patterns that empirically link to the social groups of gender and age [12,13].

Despite the elaborated literature on how gender and age stereotypes may shape evaluations, the degree to which these social processes generalize onto the digital context remains largely unexplored. In light of the exponentially augmented impact of digital platforms, knowing how social biases operate in these new domains is of major importance. This raises the question of whether women and older people are less likely to influence others online, and how specific behaviors inherent to these social groups may contribute to such biases. The present study aimed at investigating how language typically linked to gender and age relate to influence of talks shared online. Is online influence governed by the same rules as influence in offline settings, i.e.; are male features linked to more influence? Or do digital platforms eventually represent new contexts that afford different realities and rewarding opportunities for female features? In the following, we briefly review the literature on gender and age stereotypes that will lay the ground for our assumptions in the digital context.

## Gender and age as social impact factors

Two basic dimensions of social perception–warmth and competence–provide a generic framework along which most evaluations about social groups occur [4,14,15]. When forming judgments, people characterize each other by liking, i.e.; warmth, and by respecting, i.e.; competence [4]. Despite societal change and shifts in gender roles, the deeply ingrained hierarchical element of gender stereotypes has remained, often leading to men being perceived as more competent and higher status than women [6,16–18]. The traditional image of women as warm (rather than competent) may even affect high status women: As an example, female professors reported to receive more special favor requests from students, reflecting students expectations of women being"nurturing"[19].

Oftentimes, people are penalized for the display of counter-stereotypical behavior, e.g. for women to show dominance or assertiveness [8], or for men to show so-called "weak" behavior, e.g., for male leaders to seek help [9]. For women in powerful positions, e.g. female leaders, this may favor more negative evaluations due to a perceived incongruence between their role and their gender [4,5,20]. Similarly, researchers observed particular benefits of gender role-congruent behavior. As an example, female physicians received more favorable evaluations when they interacted in a warm, female-typical manner–especially when additional external cues (e.g. white coats) helped to underline their authority [21].

In a similar vein, the same type of behavior may be evaluated differently depending on whether it is displayed by a man or a woman [22,23]. The expression of anger–stereotypically masculine behavior–has been linked to lower ascribed status for angry female, but not for angry male professionals, regardless of their actual status [24]. Similar examples have been documented for the use of humor in professional settings, which may be costly in terms of ascribed status for women, but not for men [25].

To sum up, evaluations are often gendered, and masculinity is typically more closely linked to the perception of competence and high status. At the same time, the perceived congruence between the behavior and enacting person's gender role may affect evaluative outcomes, with role-congruent behavior often leading to favorable evaluations. It is important to note that most studies in this field, however, rely on a traditional unidimensional perspective on gender, in which femininity and masculinity are mutually exclusive, a framework that has been questioned in the past [26,27].

Similar dynamics as for women may be observed for older people, who are commonly perceived as higher in warmth, but lower in competence and status than younger people [4,28–30]. In the work domain, older people have been perceived as less competent [e.g.; 31,32]. Similarly in the academic context, student evaluations of faculty tended to be more positive for young, male faculty members [33]. Since older age is linked to the perception of lower status and power in similar ways as female gender, older women may be faced with double jeopardy–a phenomenon that has been referred to as "gendered ageism" [34].

Age stereotypes are, however, heterogeneous. Despite dominant negative beliefs about older people on the competence dimension, positive beliefs about older people include greater wisdom and story-telling skills [30,35]. In fact, verbal performance is usually spared from age-linked cognitive decline [36] and older people often communicate to teach younger generations [30]. It may then not be surprising that older people's communication style may contribute to the impression of their greater wisdom [30,37].

In conclusion, male gender and younger age have consistently been linked to the perception of higher competence and status, suggesting a general benefit for male and younger people in offline social evaluation and influence. The digital era has brought new possibilities to create visibility and impact at a never before seen level. Our understanding of what drives influence in the realm of online communication is, however, limited thus far. Digital communication platforms target at the popularization of science and knowledge, and speakers in these formats often adhere to an informal, entertaining style to present their ideas to a broad audience [e.g., 7]. Past research suggests that women are underrepresented on these platforms [38], and that female gender may constrain the impact of content shared online [1,7]. This raises the question of whether influence and social evaluations in the digital age are governed by the same rules as traditional settings of social evaluation: Is influence driven by features of power and masculinity? Or does the more informal setting of these new communication contexts provide new opportunities in which features of warmth and femininity are rewarded with higher impact?

## Social perception in the digital age: Stereotypes and language

Perhaps the most prominent example of a digital communication platform is TED–"Technology, Entertainment, and Design" (https://www.ted.com). TED's mission is to provide a powerful platform to spread ideas with a wide audience. At TED conferences, academics, entrepreneurs, artists and a variety of other individuals give short talks about their area of expertise. Originally launched as a small conference, TED achieved world-wide success as it began to host videos of the talks on its own website. TED Talks now cover a wide range of topics ranging from science to business and global issues, and the talks shared online have been watched over a billion times [1,7].

On the website, users can interact with the talks, e.g. by rating or commenting them, and so-called *altmetrics* [alternative metrics; 39,40] provide measures of the talks' impact, such as how many times they have been watched or rated. In line with prevalent social group stereotypes [4], previous research indicated that TED Talks' impact may differ as a function of speakers' gender–both in terms of quantity and quality [1,7]. More specifically, female speakers' TED Talks received fewer views and likes on YouTube than male speakers' talks [7], suggesting that male speakers might be in the more powerful position for impact generation. At the same time, TED Talks given by female speakers elicited more emotional discussions online: Comments on female speakers' talks were more positive and more negative (rather than neutral) compared to male speakers' talks [1].

While studies like these provide a preliminary understanding of how influence in the digital age might be affected by social processes, what remains unanswered are the specific *behaviors* that may contribute to biased evaluations and influence. One promising channel to look at in digital communication is verbal behavior. Studies that specifically examined behaviors as salient features of social roles are scarce; however, a recent laboratory-based experiment showed that evaluations were not biased in terms of gender or age when people rated *silent* excerpts of TED Talks videos [41]. It thus seems convincing that differences in the way speakers communicate their ideas may contribute to their talks' impact.

Language is crucial for shaping the impact of a message and content shared online. In social media, an analytical rather than narrative, informal or story-telling like communication styles has been linked to greater online influence [42,43]. In online pet advertisements for example, profiles of successfully adopted pets were characterized by more complex descriptions and fewer social references compared to profiles of unadopted pets [42]. Similarly, grant proposals written in complex writing styles predicted higher funding magnitude [10]. This adds to research demonstrating that analytical thinking styles are rewarded in academic contexts [10,44]. In addition, a more abstract language style has been linked to the perception of higher power in a variety of laboratory and naturalistic experiments [45]. All in all, these examples reveal insights into how complex and fact-oriented rather than personal or narrative language styles may be a successful persuasion mean both in offline and online settings.

In light of the well-documented disadvantages of women and older people in social influence, this brings up the question of how language might contribute to these differences. In fact, people's word use differs as a function of their social groups and related characteristics, e.g. their gender and age [12,13,37,46]. Abstract and analytical language–language features with greater persuasive power–are more commonly displayed by men than by women [38,46,47]. Male-typical language seems to be more concerned with references to facts and "the big picture". Conversely, female-typical language has been described as more narrative, personal, social and emotional; women tend to refer more to themselves and to other people [38,46,48–50]. While these language features have previously been linked to lower persuasive power and status [42,43,51,52], language features commonly displayed by women may at the

same time convey more psychological closeness and authenticity [47,53]. This is in line with findings suggesting that women refer more to affiliative topics in social media [54]. Rather unexpectedly, the same study found that female users of social media also used slightly more assertive language than male users.

In addition to gender, language use differs between people of different ages [37]. Language styles can thus be thought of as implicit markers of gender and age that may shape evaluative outcomes [11,55]. Compared to younger people, older people tend to show higher emotional positivity and cognitive complexity [37,56 but see 57 for contextual differences], fewer self- and more references to other people [13,37], and more certainty and fluency in their language [58]. Language commonly used by older people thus seems to simultaneously convey warmth, competence and wisdom [4,59], as well as higher status [51,52]. In contrast to common negative beliefs tied to chronological age [30,60], we may expect that more senior language conveys more positive attributes commonly linked to aging, such as wisdom [30,35].

Despite the crucial role of language in social perception, the question of how verbal behavior, i.e.; gendered and age-linked language styles, link to social evaluation has to our knowledge not yet received much scientific attention. In the present study, we aimed at filling this gap by investigating how speaker's gender and age, and particularly prototypical language markers of gender and age, predict online influence.

## The present study

Within the scope of the present study, we focused on language use patterns commonly associated with gender and age to examine their link with quantity and quality of impact, that is the number of talk views and the positivity of talk ratings on the TED website. Our aims were to develop a preliminary understanding of how language use–as a behavioral manifestation of speaker's genders and age–relate to social evaluation and impact, beyond speakers' identified gender and chronological age. In other words, we considered prototypical verbal behavior as a potential mechanism to activate social processes in evaluations beyond other cues, such as visual displays of gender or age that might provoke stereotyping. Furthermore, we were interested in the possible interplay between chronological age, identified gender, and gender- and age-linked language displays. Due to the novelty of the question and suggested partial overlaps between gender- and age-linked language styles [e.g.; 50], we investigated the effects separately for each of the two language signatures.

The main aim of TED Talks is to exert social influence, but at the same time to communicate ideas and knowledge in a simple and engaging way. The question of whether influence operates in comparable ways in this novel communication space as in offline interactions has not yet been studied. Is a more instrumental and complex male-typical language style predictive of TED Talk impact, or rather a simpler and more personally engaging female-typical language style? And are these associations the same for male and for female speakers? We addressed these questions in two competing hypotheses, the *male over female-hypothesis* versus the *congruent is prudent-hypothesis*.

Due to the well-documented male advantage in social influence [5,6], we expected a general advantage of male-typical language style in terms of talk impact (*male over female-hypothesis*). We assumed that this might be the case for women in particular, namely that a male language style might help them overcome the ascribed lower status typically associated with their gender.

In a competing hypothesis, we followed argumentations that gender role-congruent behavior is socially rewarded [61] and expected positive effects on TED Talk impact if speakers use language that conforms with their own gender, i.e.; if female speakers use female-typical and male speakers male-typical language (*congruent is prudent-hypothesis*).

In addition to gender-linked language style, we examined the links between *age-linked language styles* and talk impact. In contrast to chronological age that is often linked to the perception of lower competence and status [4], the specific ways in which older people use language may convey positive features, such as wisdom or high status [37,51]. We expected differential effects of speakers' chronological age and age-linked language style on TED Talk impact, namely that a more senior language style links to greater talk impact.

**Research questions and hypotheses.** In sum, we investigated how speakers' gender, age and gender- and age-linked language styles are linked to talk impact both in quantity (number of views) and quality (proportion of positive and negative ratings). While our hypotheses regarding impact quality were pre-registered (osf.io/jvp6r and osf.io/7ksvx), our analyses of talk views were not pre-registered. The second research question was preregistered as an addendum to the first one, before running the analyses.

**Question 1. Do TED Speakers' gender and gender-linked language predict talk impact?.** First, we investigated the main effects of speaker's gender on talk impact and expected that male speakers' talks receive higher proportions of positive ratings than female speakers' talks. Secondly, we aimed to capture the unique effects of gendered language style on talk impact, while taking into account that effects of language style may differ depending on the speaker's gender (as indicated by an interaction effect speaker's gender × gendered language style). We investigated the following two competing hypotheses for gendered language style on talk ratings:

a. Male-typical language links to more positive ratings in general and in particular for female speakers *(male over female-hypothesis)*.

b. Gender congruent language use, i.e., female speakers with female-typical language for female speakers and male-typical language for male speakers, links to more positive ratings *(congruent is prudent-hypothesis)*.

**Question 2. Do TED Speakers' chronological age and age-linked language style predict talk impact?.** We expected opposing associations for speakers' chronological age and age-linked language style with talk ratings. Specifically, we expected that speakers' older chronological age links to less positive ratings and that speakers' more senior language style links to more positive talk ratings. For chronological age, we more specifically expected older speakers' talks to receive fewer positive and more negative talk ratings than young- to middle-aged speakers' talks. We further tested for possible interaction effects between speakers' chronological age, age-linked language style and gender, and expected the associations between age and talk ratings to be moderated by speaker's gender ("gendered ageism"; e.g. [34]).

## Method

We collected transcripts of English TED Talks along with their impact (talk ratings, number of views) and other informative measures (e.g.; date the talk was given) from the official TED Talks website in March, 2018. The data collection method complied with the terms and conditions for the TED website (see: https://www.ted.com/about/our-organization/our-policies-terms/ted-com-terms-of-use). Data was collected as part of a larger project that examined psychological adaptation in translations [38]. We started from the sample of $N = 1,647$ TED Talks used in [38], which only included talks given by single speakers that were no live other artistic performances, and we had to exclude seven talks for which information about talk ratings and views was missing. Since it was important to control for each speaker's age in our analyses, we only included talks by speakers for whom information on their age was available from internet searches, which was the case for $N = 1,095$ speakers that formed our final sample. Overall, our final sample included TED Talks that had been delivered between 1990 and 2017.

## Measures

We briefly describe the impact measures along any other measures used in our analyses here.

**Quantity of talk impact.**   As measures of overall talk popularity and impact, we used the total number of views any given talk received, that is, the number of clicks the talks had on the TED website. On average, TED Talks were viewed 2,073,083 ($SD$ = 3,607,560) times.

**Quality of talk impact: Positive and negative talk ratings.**   On the TED website, users have the possibility to rate talks by selecting 3 adjectives out of a given list of 14 ratings. At the very end of each video, TED's instruction to rate the talk appears as follows: *"How would you describe this talk? Tell us by choosing up to three words. (If you choose just one, it will count three times.)" "Beautiful", "courageous", "ingenious", "informative", "persuasive", "funny", "fascinating", "inspiring", "longwinded", "unconvincing", "obnoxious", "OK", "jaw-dropping"and "confusing"*.

In our analyses, we focused on aggregated percentages of positive and negative ratings, as well as percentages of each of these ratings individually. The individual ratings can be thought of as different facets of positivity or negativity. We considered *beautiful*, *courageous*, *ingenious*, *informative*, *persuasive*, *funny*, *fascinating*, *inspiring* and *jaw-dropping* as positive ratings, and *longwinded*, *unconvincing*, *obnoxious* and *confusing* as negative ratings respectively. We did not include *"ok"* in either of the aggregated scores, as it could be thought of as a neutral category. While the aggregated negative scale showed acceptable internal consistency (Cronbach's Alpha = 0.76), the aggregated positive scale was not internally consistent (Cronbach's Alpha = -4.84). This indicates that the different positive ratings are not empirically covarying and that the aggregated positivity scale should be interpreted with caution. On average, TED Talks received 2,990.99 ($SD$ = 5,598.32) ratings, out of which 87.79% ($SD$ = 10.41) were positive ratings and 7.95% ($SD$ = 8.33) negative ratings.

**Speaker's gender.**   TED Speakers have a personal profile on the TED website which includes their short biography. We used the gender coding of a previous study [38], in which the genders of speakers were coded based on the videos and information provided on their public profiles. Transgender individuals were coded in terms of the gender they identified with, i.e. personal pronoun used in the profile descriptions ($N$ = 4 in our sample), conforming with recommended practices on gender identity measures [62].

**Speaker's age.**   Information about chronological age of TED Speakers was obtained from google searches about the speakers as well as a publicly available database (https://www.wikidata.org/wiki/Wikidata:TED/TED_speakers). If available, we collected information about the speakers' exact birthdates, or the year they were born. If only the year of their birth was known, date of birth was coded as"January 1"of the given year. In other words, if only the year, but not the exact date of birth was known, we took the year of birth as reference to infer their age at the time the talk was given. In case the year of birth was unknown, but information about their age at a certain point in time was available (e.g. a newspaper article from 2015 indicating that the speaker was 40 years old at that time), we coded the later of the two possibilities as their year of birth. In the example above, we would have taken "January 1, 1975" as date of birth. We then calculated chronological age [years] by subtracting their date of birth from the date the talk was given. TED Speakers' mean age of was 47.29 years ($SD$ = 12.82), with the youngest speaker being 12, and the oldest speaker 94 years old.

**Gender-linked language.**   We used previously validated dictionaries [12] to quantify speakers' gender- and age-linked language use from the transcripts. The dictionaries contain weighted lists of words that have previously been successful in predicting authors' gender and age from text. Both lexica have been widely used in research on social media [63–66], and the gender lexicon has been shown to achieve 91.9% prediction accuracy in determining gender from language [12].

We consider the gender and age scores as measures for gender- and age-style prototypical-ity in language and use them to predict talk ratings and views in order to infer the role of gen-der- and age-linked language in stereotypical social evaluations. Negative values on the gender score refer to a more male-typical language style, and positive values to a more female-typical language style. On average, TED Speakers language was slightly more male-typical ($M$ = -0.32, $SD$ = 1.69), and the majority of speakers (66.9%) had a language style that conformed with the typical language style of their own gender (60.9% of female speakers had a female-typical lan-guage style, and 69.7% of male speakers had a male-typical language style).

**Age-linked language.**   Analogously, the age score can be thought of as a behavioral mani-festation of age in language, with higher values indicating language use that is more typical for older people. Previously, this score has been shown to predict age based on language with an accuracy of $r$ = .83 [12]. Speakers' age-linked language style and chronological age correlated at Pearson's $r$ = .23 ($p <$ .001) in our sample. Overall, TED Speakers language was classified as rather "young" based on the lexicon ($M$ = 28.08 years, $SD$ = 5.39) compared to their chrono-logical age ($M$ = 47.29 years, $SD$ = 12.82). The two language measures for gender and age cor-related at $r$ = .13 ($p <$ .001).

**Speakers' academic status.**   In addition to speaker's gender and age, we included their academic status as a control variable in our analyses. This procedure was informed by previous findings showing that public trust in scientists is high for researchers in academia [67], and that TED presenters' academic status links to their talk impact [7]. Since the present study's primary interest lay in stylistic aspects of TED Talks, this can also be seen as a way to control for content of TED Talks. Following the procedure proposed by [7], we coded speakers as "academic", if they had earned a doctoral degree, and if they were affiliated with an academic institution at the time they gave their TED talk. Academic institutions were defined as degree-granting institutions offering full programs both at the undergraduate and graduate level. We coded individuals as non-academic if their doctoral degree was still in progress or had never been obtained. This allowed us to control for speakers' academic status in a way that conforms to prior research in the context of TED Talks (see [7]).

A sample description a long with descriptive information on gender- and age-linked lan-guage style is provided in Table 1. Summary information for talk views and ratings is presented in Table 2 (see also Tables A and B in S1 File for inter-correlations and additional information on talk ratings).

**Table 1. Descriptive summary of the sample.**

|  | *N* Speakers | Speaker's Age *M (SD)* | Gender-Linked Language *M (SD)* | Age-Linked Language *M (SD)* |
|---|---|---|---|---|
| Female Speakers | 348 (31.8%) | 44.10 (12.95) | 0.50 (1.87) | 29.19 (5.58) |
| Male Speakers | 747 (68.2%) | 48.77 (12.49) | -0.69 (1.46) | 27.56 (5.22) |
| Total Speakers | 1,095 (100%) | 47.29 (12.82) | -0.32 (1.69) | 28.08 (5.39) |
| Female Academics | 62 (28.2%) | 45.73 (11.02) | 0.36 (1.78) | 28.78 (5.92) |
| Male Academics | 158 (71.8%) | 51.79 (11.16) | -0.89 (1.40) | 27.77 (4.91) |
| Total Academics | 220 (20.1%) | 50.08 (11.43) | -0.54 (1.61) | 28.06 (5.22) |
| Female Non-Academics | 286 (32.7%) | 43.75 (13.33) | 0.53 (1.89) | 29.28 (5.51) |
| Male Non-Academics | 589 (67.3%) | 47.96 (12.72) | -0.64 (1.47) | 27.50 (5.30) |
| Total Non-Academics | 875 (79.9%) | 46.59 (13.06) | -0.26 (1.71) | 28.09 (5.43) |

*Note*. *M* = Mean, *SD* = Standard deviation. Gender-linked language: Negative values refer to a more male-typical, positive values to a more female-typical language style; age-linked language: Higher values refer to a more senior language style [c.f.; 12].

Table 2. Summary information for TED talk altmetrics.

| Total speakers N = 1,095 | | Male speakers N = 747 | | | Female speakers N = 348 | | |
|---|---|---|---|---|---|---|---|
| | | Total male N = 747 | Academics N = 158 | Non-academics N = 589 | Total female N = 348 | Academics N = 62 | Non-academics N = 286 |
| | M (SD) | M (SD) | M (SD) | M (SD) | M (SD) | M (SD) | M (SD) |
| Total views | 2,073,083 (3,607,560) | 2,025,416 (3,406,197) | 1,841,940 (2,461,266) | 2,074,633 (3,618,123) | 2,175,404 (4,009,097) | 3,527,415 (7,364,713) | 1,882,311 (2,734,319) |
| Views adjusted for time online | 1,271.20 (2,306.05) | 1,149.73 (2,345.41) | 1,025.15 (2,212.81) | 1,183.15 (2,380.42) | 1,531.92 (2,199.92) | 1,962.50 (3,322.89) | 1,438.58 (1,865.51) |
| Time online (video) [days] | 2,511.12 (1,386.35) | 2,699.81 (1,396.43) | 2,720.65 (1,303.54) | 2,694.22 (1,421.32) | 2,106.08 (1,275.18) | 2,059.45 (1,140.19) | 2,116.19 (1,304.22) |
| Total positive ratings % | 87.79 (10.41) | 87.51 (10.65) | 86.87 (10.50) | 87.67 (10.69) | 88.39 (9.86) | 88.99 (7.79) | 88.26 (10.27) |
| Inspiring % | 19.35 (10.63) | 18.62 (10.54) | 14.07 (8.75) | 19.84 (10.65) | 20.92 (10.68) | 15.25 (9.48) | 22.15 (10.54) |
| Beautiful % | 7.18 (7.60) | 6.36 (7.55) | 3.64 (3.83) | 7.09 (8.11) | 8.95 (7.41) | 5.19 (3.99) | 9.77 (7.72) |
| Ingenious % | 6.26 (5.93) | 7.01 (6.19) | 7.44 (6.17) | 6.90 (6.19) | 4.63 (4.98) | 4.35 (3.56) | 4.69 (5.24) |
| Courageous % | 7.06 (7.80) | 5.81 (6.73) | 3.39 (4.13) | 6.45 (7.13) | 9.76 (9.16) | 5.00 (6.69) | 10.79 (9.31) |
| Jaw-dropping % | 4.70 (5.38) | 5.12 (5.75) | 5.34 (5.98) | 5.06 (5.69) | 3.81 (4.37) | 3.80 (3.50) | 3.81 (4.54) |
| Fascinating % | 13.07 (7.38) | 13.81 (7.36) | 17.34 (8.25) | 12.87 (6.80) | 11.49 (7.18) | 17.14 (8.68) | 10.26 (6.19) |
| Informative % | 16.15 (9.98) | 16.26 (9.79) | 21.74 (8.78) | 14.79 (9.53) | 15.91 (10.39) | 25.67 (9.09) | 13.79 (9.41) |
| Funny % | 5.04 (8.77) | 5.29 (9.17) | 3.45 (5.15) | 5.79 (9.92) | 4.50 (7.85) | 3.48 (6.07) | 4.72 (8.18) |
| Persuasive % | 8.97 (6.90) | 9.23 (7.14) | 10.47 (7.15) | 8.89 (7.11) | 8.41 (6.33) | 9.11 (4.79) | 8.26 (6.61) |
| Total negative ratings % | 7.95 (8.33) | 8.18 (8.47) | 8.79 (8.58) | 8.02 (8.45) | 7.44 (8.00) | 6.77 (6.06) | 7.59 (8.37) |
| Obnoxious % | 1.45 (2.17) | 1.42 (2.17) | 1.16 (1.27) | 1.49 (2.35) | 1.51 (2.18) | 1.18 (1.36) | 1.58 (2.31) |
| Longwinded % | 2.24 (2.76) | 2.37 (2.93) | 2.58 (2.89) | 2.32 (2.94) | 1.95 (2.33) | 1.81 (1.54) | 1.98 (2.47) |
| Unconvincing % | 3.00 (3.83) | 3.07 (3.87) | 3.42 (4.27) | 2.98 (3.76) | 2.86 (3.72) | 2.65 (3.28) | 2.90 (3.82) |
| Confusing % | 1.26 (1.68) | 1.32 (1.80) | 1.63 (1.84) | 1.24 (1.79) | 1.13 (1.37) | 1.13 (1.29) | 1.13 (1.39) |
| OK % | 4.26 (3.15) | 4.31 (3.22) | 4.34 (2.68) | 4.30 (3.35) | 4.16 (3.00) | 4.24 (2.44) | 4.15 (3.11) |

*Note.* M = Mean, SD = Standard deviation. Time online refers to the number of days the talk video had been on the TED website prior to data collection.

Positive ratings = aggregated score of all positive ratings, negative ratings = aggregated score of all negative ratings, "ok" was not considered in the aggregated ratings.

For more information on ratings (counts), see Table A in S1 File.

## Statistical analyses

To address our research questions, we computed series of beta and quantile regressions. Our data and data analysis scripts are available on the Open Science Framework: https://osf.io/qkm6u/. For the analyses on *quantity* of talk impact, we performed two series of quantile regressions to predict the number of views, separately for gender- and age-linked language style. This modelling method allows to adequately deal with the particular distribution of the data; i.e.; the great variability in talk views (min = 185,525; max = 50,458,477). Quantile regressions are robust against outliers and allow for a more comprehensive analysis of the relationship between variables by enabling estimation of conditional quantiles of the dependent variable rather than the mean, while taking into account the whole sample [68]. It seemed plausible that effects would differ depending on the quantile, e.g. be more or less pronounced among the top versus less impactful talks. We used the package "quantreg" in R [69] to estimate the relationship for 10%, 25%, 50%, 75%, and 90% quantiles of talk views, thus estimating the effects separately for talks of average and high/low influence.

For the analyses on *quality* of talk impact, our dependent variables were aggregated positive ratings (%), aggregated negative ratings (%), and all 14 talk ratings (%) individually. We

computed two series of beta regressions with logit-links using the R-package "betareg" [70] to predict proportions of positive and negative ratings from gender- and age-linked language style. Beta regressions are the state-of-the-art modelling procedure for continuous outcomes that are bounded within intervals of [0,1], such as percentages [71]. The effects in beta regressions can be expressed as odds ratios.

In the models of our first research question about gender and gender-linked language style, the model predictors were the main effects of speaker's gender and gender-linked language use, as well as their interaction term gender × language style. Controlling for speaker's gender in the analysis can be seen as a conservative approach to examine the unique effect of gender-linked language style on talk impact beyond speaker's gender. The interaction term speaker's gender × gender-linked language use further allowed us to test whether effects of gender-linked language style on talk impact differed between male and female speakers. Since talk impact might also depend on speaker demographics other than gender, or by how long the video had been available on the website, we included the time the talk had been online [days between data collection and date talk was given], speaker's age and academic status as control variables. For the models on impact *quality* (positive and negative rating proportions), we additionally controlled for the total number of talk ratings received.

Since we expected positive links between male gender and positive talk ratings, we used female gender as reference category. The odds ratios for speaker's gender therefore represent the likelihood to receive rating types when presenting as male speaker. In contrast, higher scores on the gendered language metric [12] reflect more female language style. We retained this scoring direction in order to facilitate comparisons with earlier studies using this metric. In order to enable a more meaningful interpretation, we z-transformed the gendered language score prior to inclusion in our models. A one unit increase in the gendered language score thus refers to a one standard deviation increase towards the female-typical direction. Odds ratios thus represent the increased likelihood to receive rating types for a one standard deviation increase in female-typical language style.

Similarly, in the models of our second research question about age and age-linked language style, the model predictors were the linear and quadratic effects of speaker's age and of age-linked language style. Likewise, this allowed us to test whether age-linked language style predicts talk ratings above and beyond the effect of speaker's chronological age. Moreover, including quadratic effects enabled us to examine whether age effects differ across time points in the lifespan, such as whether positive ratings peak at middle adulthood. Following recommendations to remove non-essential correlation between the linear and quadratic effects [72], both chronological age, and the language score of age were mean-centered, thus placing the zero value as the mean age and age score of the sample within the range of the data.

We further included interaction effects of speaker's age × speaker's gender, speaker's age squared × speaker's gender, as well as age-linked language style × speaker's gender, age-linked language style squared × speaker's gender to control for the possibility that age differences in talk impact were different for male and female speakers (i.e.; gendered ageism effects). Again, we controlled for speaker's gender and academic status, as well as the time the talk had been online. For the models on positive and negative ratings, we also controlled for the total number of talk ratings received.

Conforming to current methodological recommendations [73], we report exact *p*-values without adjusting for multiple testing. In order to offer interpretations beyond *p*-values, we further report 95% confidence intervals of the estimates. Goodness-of-fit indicators of beta regressions are reported as $R^2$ in Tables G, H, K and L in S1 File.

## Results

### Question 1. Do TED Speakers' gender and gender-linked language predict talk impact?

**Gender, gender—linked language and impact quantity.**   Fig 1 illustrates the effects of gender-linked language style on talk views from quantile regressions. A more female language style was linked to greater impact of TED Talks. This was especially the case among the most popular talks, i.e.; talks with a lot of views, as indicated by the steeper slopes among the highest quantiles (see Fig 1, and also Table C in S1 File). A more female language style predicted more views in all quantiles ($p < .05$) except the 10% quantile. This unexpected advantageous effect of female language style was in contrast to the effects of speaker's gender, according to which male speakers' talks received more views than female speakers' talks ($p < .05$)–this relationship held for talks of average and extremely high popularity, i.e., 25%, 50%, and 90% quantiles of views (see Table C in S1 File for more details).

The size of the effects of female language on impact quantity can be exemplified as follows: A one standard deviation increase in female language style among the extremely often viewed talks (i.e.; 90% quantile) was linked to 723,286.42 (343,429.37) more talk views. A more female-typical language style was therefore linked to greater talk impact in terms of quantity, and these beneficial effects were more pronounced the more popular the talks were. A more female, and thus more narrative way of communicating may fit with TED's scope to convey complex ideas in a concise and engaging manner. Similar relationships were also found between gender-linked language style and *number* of talk *ratings*: Female-typical language style was linked to more talk ratings overall, particularly among the most popular talks (see Table D in S1 File for more details). Since the instruction to rate talks did not appear until the very end of each video, this suggests that talks given in a female-typical language style were more likely to be watched until the end.

**Gender, gender–linked language and impact quality.**   The results on talk impact *quality* are presented in Figs 2–5. Figs 2 and 3 depict the effects of speaker's gender, and Figs 4 and 5 the effects of speaker's gender-linkeded language style on proportions of positive and negative talk ratings. More specifically, all figures depict odds ratios (with 95% confidence intervals), which correspond to the exponentiated estimates from beta regressions. In the figures, the meaningful effects are those for which the confidence interval does not include 1. Odds ratios > 1 mean that it is more likely for a group to receive the rating type in question, compared to the reference group, whereas odds ratios < 1 mean that it is less likely. Fig 2 depicts the likelihood for male speakers to receive the rating type in question compared to female speakers.

For details on the main results please refer to Tables G and H in S1 File. In the aggregated scores of positive and negative talk ratings, no clear pro-male bias was evident: Speaker's gender was neither associated with aggregated positivity ($p = .150$), nor with aggregated negativity ($p = .085$) of ratings. Likewise, speaker's gender-linked language style was associated with neither the aggregated positivity ($p = .276$), nor with aggregated negativity score ($p = .136$) of ratings. It must be noted that the association between gender, gender-linked language style and aggregated positivity is hard to interpret due to the low internal consistency of the aggregated positivity score.

What we did observe, however, were differences in the specific types of positive and negative talk ratings female and male speakers' talk received: Male speakers' talks received more of the positive ratings"ingenious"($B = 0.31$, 95% CI = 0.20; 0.41; $p < .001$),"jaw-dropping"($B = 0.11$, 95% CI = 0.004; 0.21; $p = .043$),"fascinating"($B = 0.15$, 95% CI = 0.06; 0.23; $p < .001$), and"funny"($B = 0.26$, 95% CI = 0.13; 0.39; $p < .001$),–ratings that seemed to imply impression, approval, or competence–but fewer of the ratings"beautiful"($B = -.025$, 95% CI = -0.36; -0.14; $p < .001$), and"courageous"($B = -0.29$, 95% CI = -0.41; -0.18; $p < .001$)–thus seemingly fewer ratings

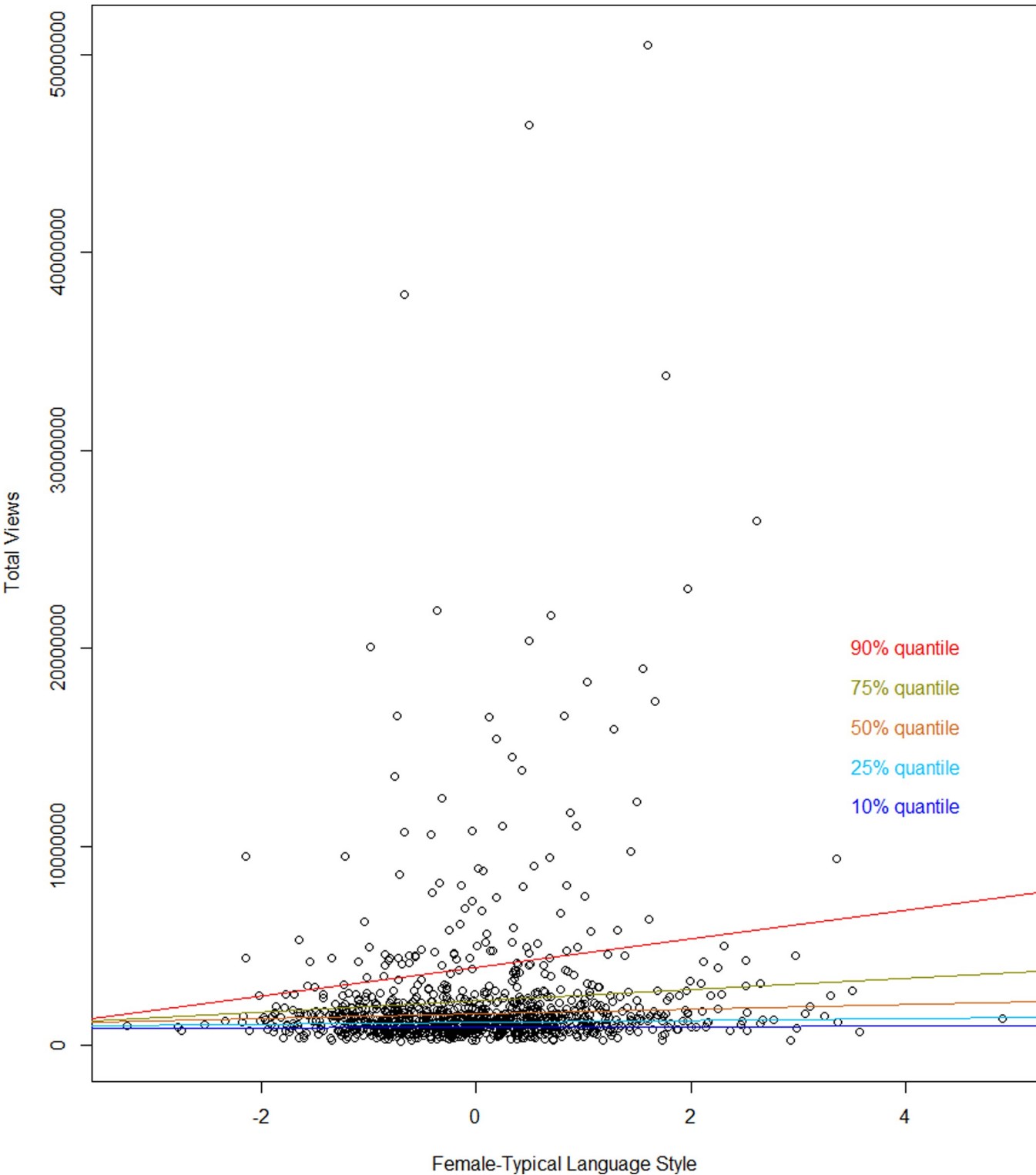

**Fig 1. Overall number of talk views by female-typical language style.** Results from quantile regressions. Fitted regression lines separately for the 10%, 25%, 50%, 75%, and 90% quantiles of the total number of views indicate more pronounced positive associations between female language style and quantitative impact among the often watched talks. Depicted are the effects of female language style while accounting for control variables (gender, age, academic status, time online; see Table C in S1 File for details).

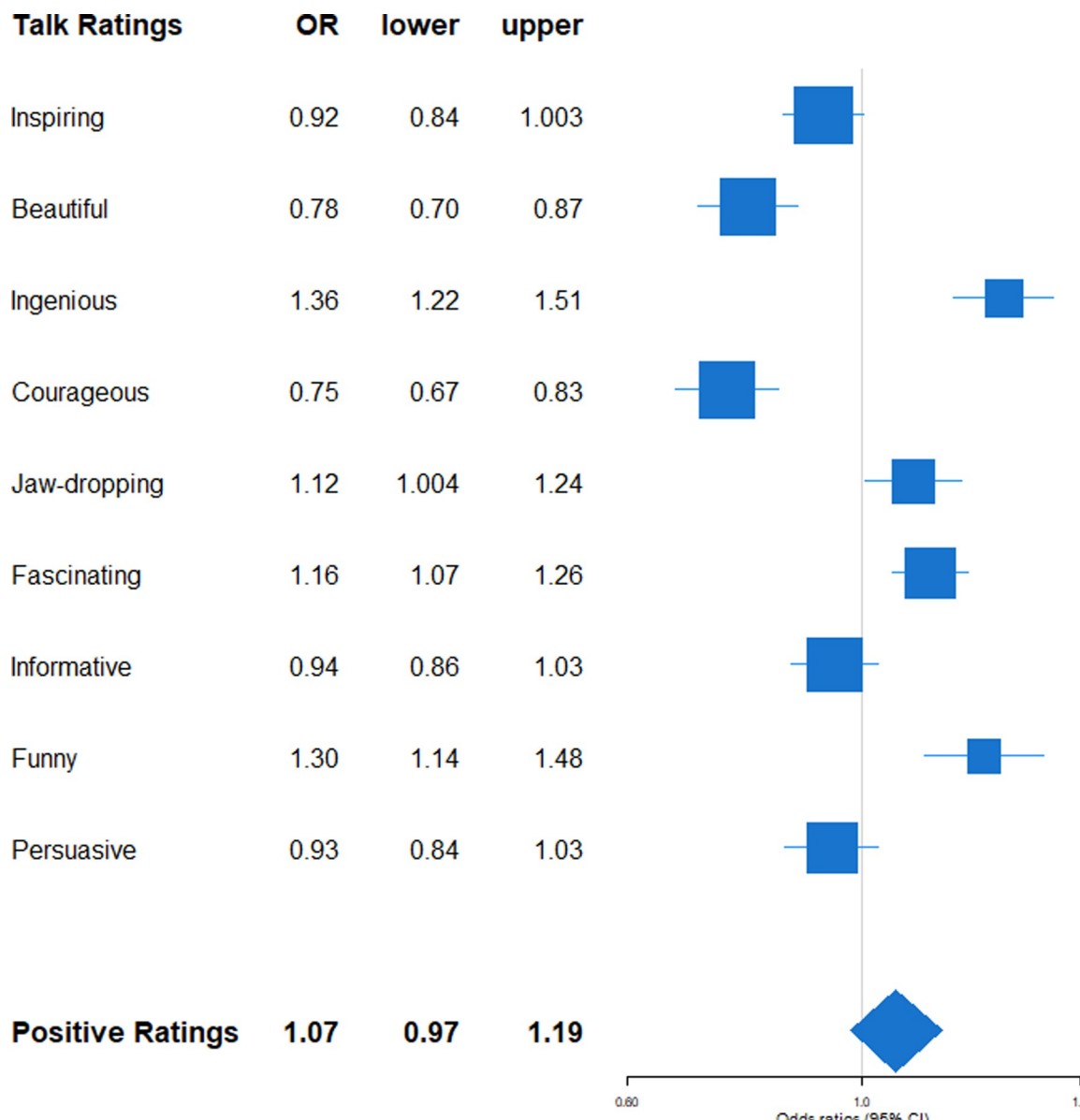

**Fig 2. Positive talk ratings by speaker's gender.** Exponentiated regression coefficients (OR: Odds ratios) and 95% confidence intervals (lower: lower bound, upper: upper bound) from beta regressions accounting for control variables (gendered language use, age, academic status, time online, number of ratings). OR > 1 indicate an increased likelihood for male speakers to receive the rating type. Positive talk ratings (boldface) refers to the aggregated score of all positive ratings.

referring to warmth or emotions (see Fig 2). Moreover, male speakers' talks received fewer of the negative ratings"obnoxious"($B$ = -0.11, 95% CI = -0.22; -0.01; $p$ = .037) and"unconvincing"($B$ = -0.11, 95% CI = -0.22; -0.001; $p$ = .049; see Fig 3), and thus seemed to earn fewer hostile attitudes and ratings referring to perceived lack of competence. It must be noted that, on average, TED Talks only received very few negative ratings (see Table 2).

More importantly however, speaker's gender-linked language style predicted talk rating types above and beyond speaker's gender: More female-typical language style predicted more of the positive ratings"beautiful"($B$ = 0.10, 95% CI = 0.03; 0.17; $p$ = .005),"courageous"($B$ = 0.10, 95% CI = 0.02; 0.17; $p$ = .010), and"funny"($B$ = 0.29, 95% CI = 0.21; 0.38; $p$ < .001), and fewer

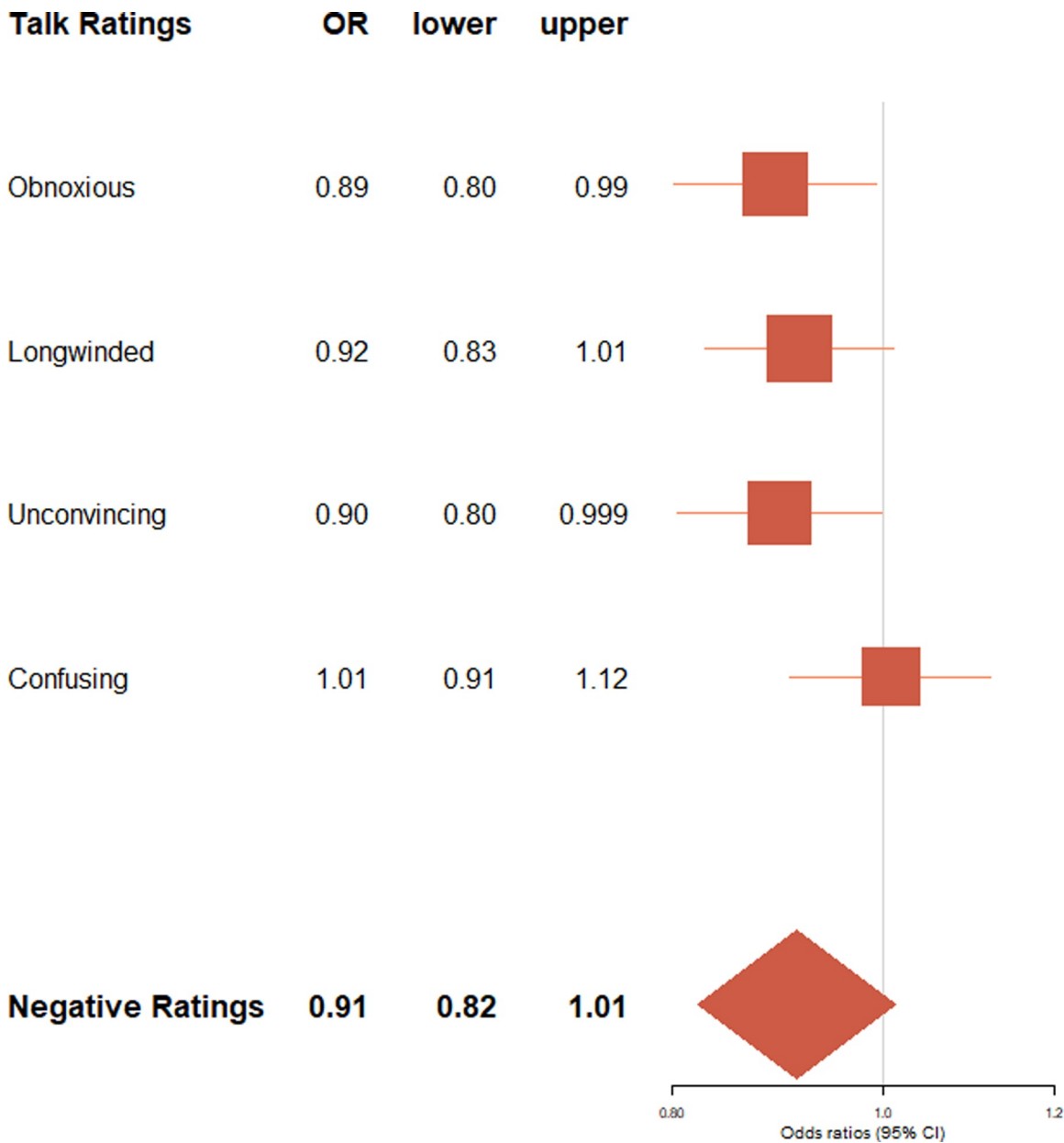

**Fig 3. Negative talk ratings by speaker's gender.** Exponentiated regression coefficients (OR: Odds ratios) and 95% confidence intervals (lower: lower bound, upper: upper bound) from beta regressions accounting for control variables (gendered language use, age, academic status, time online, number of ratings). OR > 1 indicate an increased likelihood for male speakers to receive the rating type. Negative talk ratings (boldface) refers to the aggregated score of all negative ratings.

of the positive ratings"ingenious"($B$ = -0.13, 95% CI = -0.20; -0.05; $p$ = .002),"fascinating"($B$ = -0.06, 95% CI = -0.12; -0.01; $p$ = .031),"informative"($B$ = -0.13, 95% CI = -0.20; -0.07; $p$ < .001), and"persuasive"($B$ = -0.08, 95% CI = -0.15; -0.01; $p$ = .021; see Fig 4). More female-typical language style further predicted fewer"unconvincing"ratings ($B$ = -0.09, 95% CI = -0.17; -0.01; $p$ = .021) (see Fig 5). Independently of speaker's gender, speaker's gender signature in language thus uniquely predicted different facets of positive and negative talk ratings.

Since the interpretation of effect sizes in beta regressions is not as straightforward as in other types of regressions, we report average marginal effects (AME) or average model

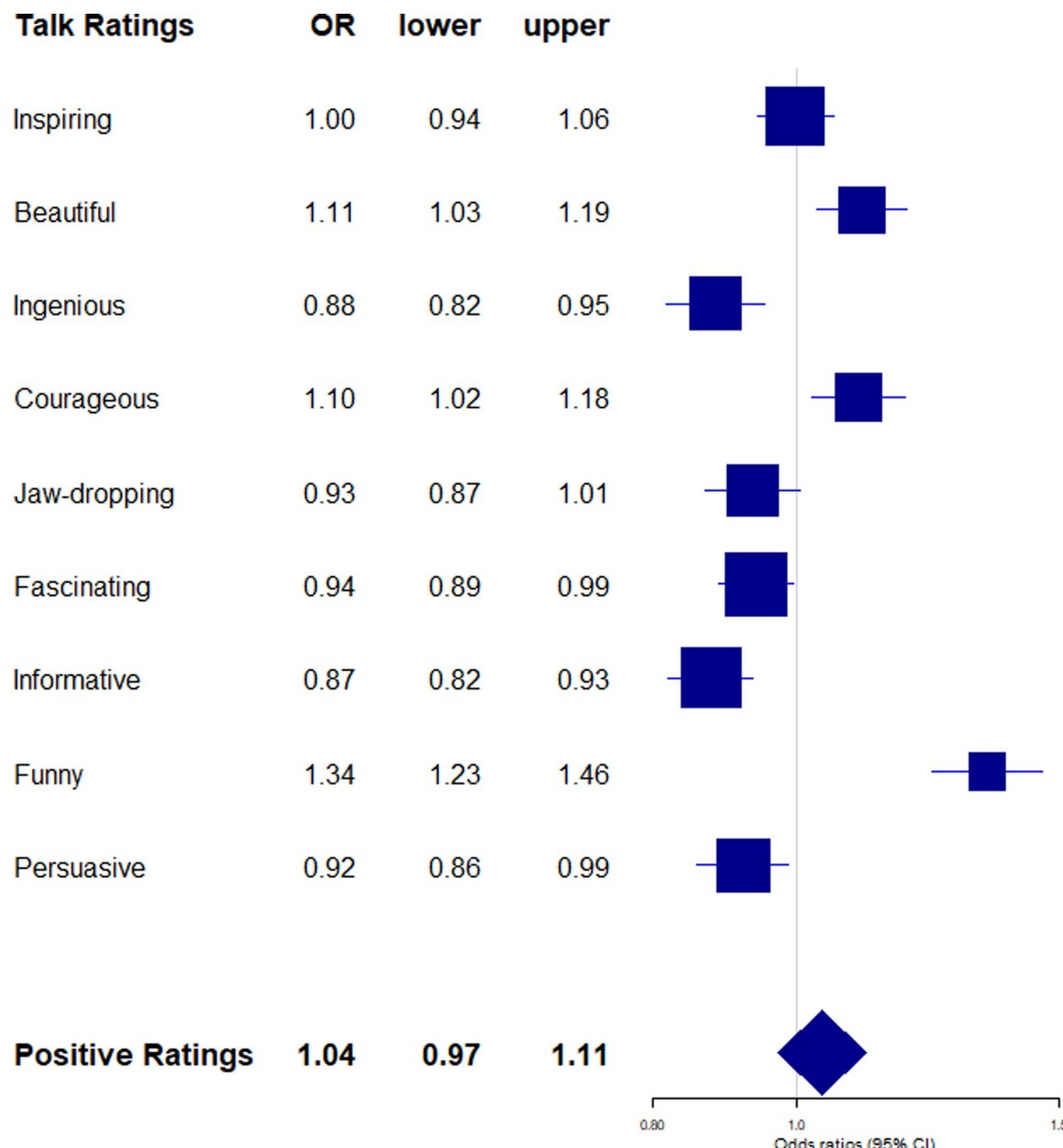

**Fig 4. Positive talk ratings by female-typical language style (z-standardized score).** Exponentiated regression coefficients (OR: Odds ratios) and 95% confidence intervals (lower: lower bound, upper: upper bound) from beta regressions accounting for control variables (gender, age, academic status, time online, number of ratings). OR > 1 indicate an increased likelihood for talks with more female language style to receive the rating type. Positive talk ratings (boldface) refers to the aggregated score of all positive ratings.

coefficients in Tables I, J, M and N in S1 File, and briefly illustrate this with an example here. The absolute size of an effect in beta regression depends on the value of the outcome. If the exponentiated coefficient of a predictor is 3, this could be a large effect (in terms of additional increase in %) if the outcome is relatively small, or it could be a small effect, if the outcome is relatively large. For gender-linked language style, the predictor used in our models was the $z$-standardized femininity score. If we take the example of"funny"ratings, on average, TED Talks had 5.04%"funny"ratings, and the AME of the femininity score is 0.014

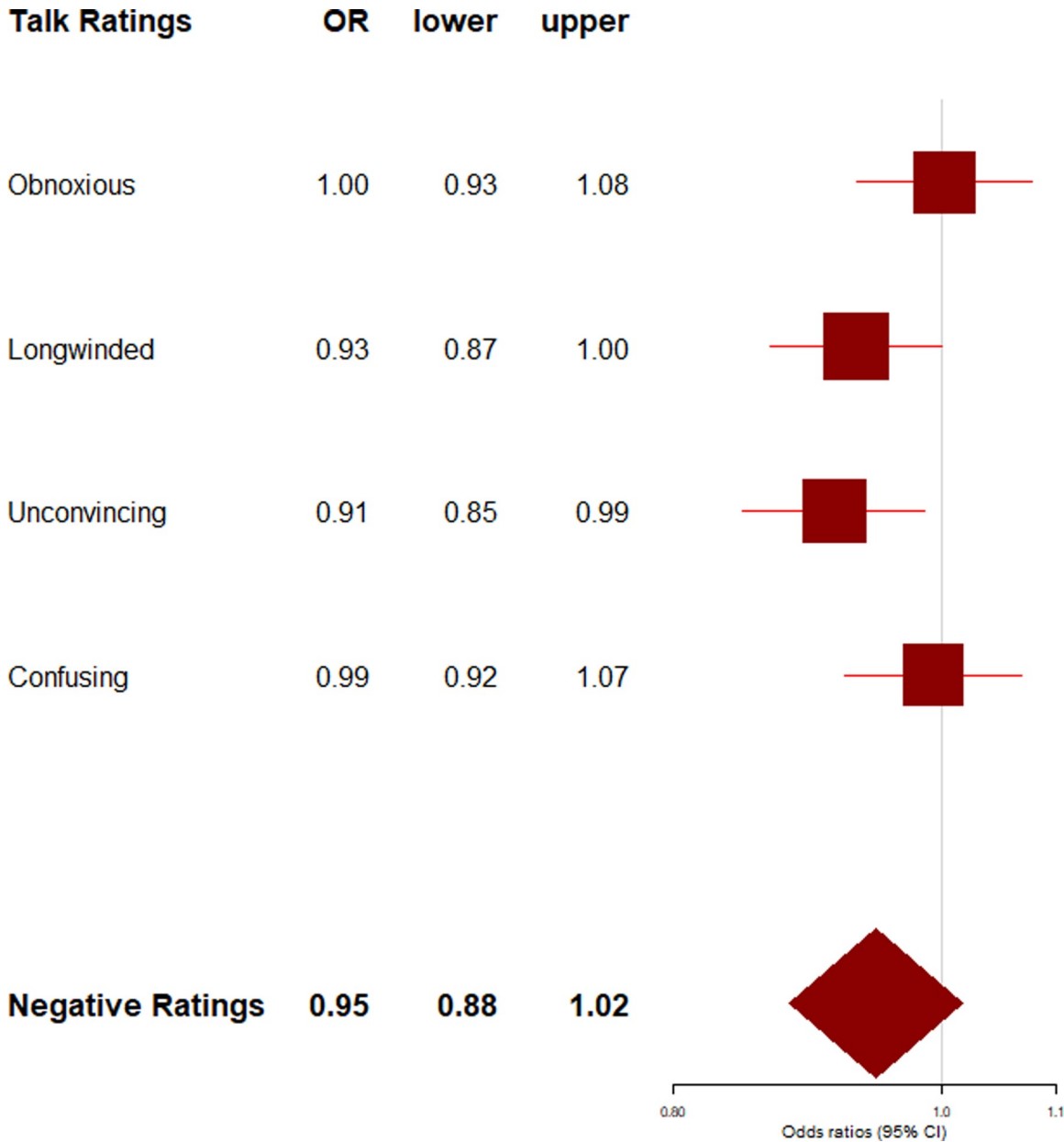

**Fig 5. Negative talk ratings by female-typical language style (*z*-standardized score).** Exponentiated regression coefficients (OR: Odds ratios) and 95% confidence intervals (lower: lower bound, upper: upper bound) from beta regressions accounting for control variables (gender, age, academic status, time online, number of ratings). OR > 1 indicate an increased likelihood for talks with more female language style to receive the rating type. Negative talk ratings (boldface) refers to the aggregated score of all negative ratings.

(see Table I in S1 File). A one standard deviation change towards a more female language style thus corresponded to a 1.4% increase in"funny"ratings, if the other model predictors were held constant. Similarly, the AME of"speaker's gender"on"courageous"was 0.021. Presenting as male gender (rather than female gender) thus, on average, linked to a 2.1% decrease in"courageous"ratings.

In sum, the results provide little support for our assumption that male gender and male-typical language relate to more positive talk impact and our *male over female-hypothesis* was

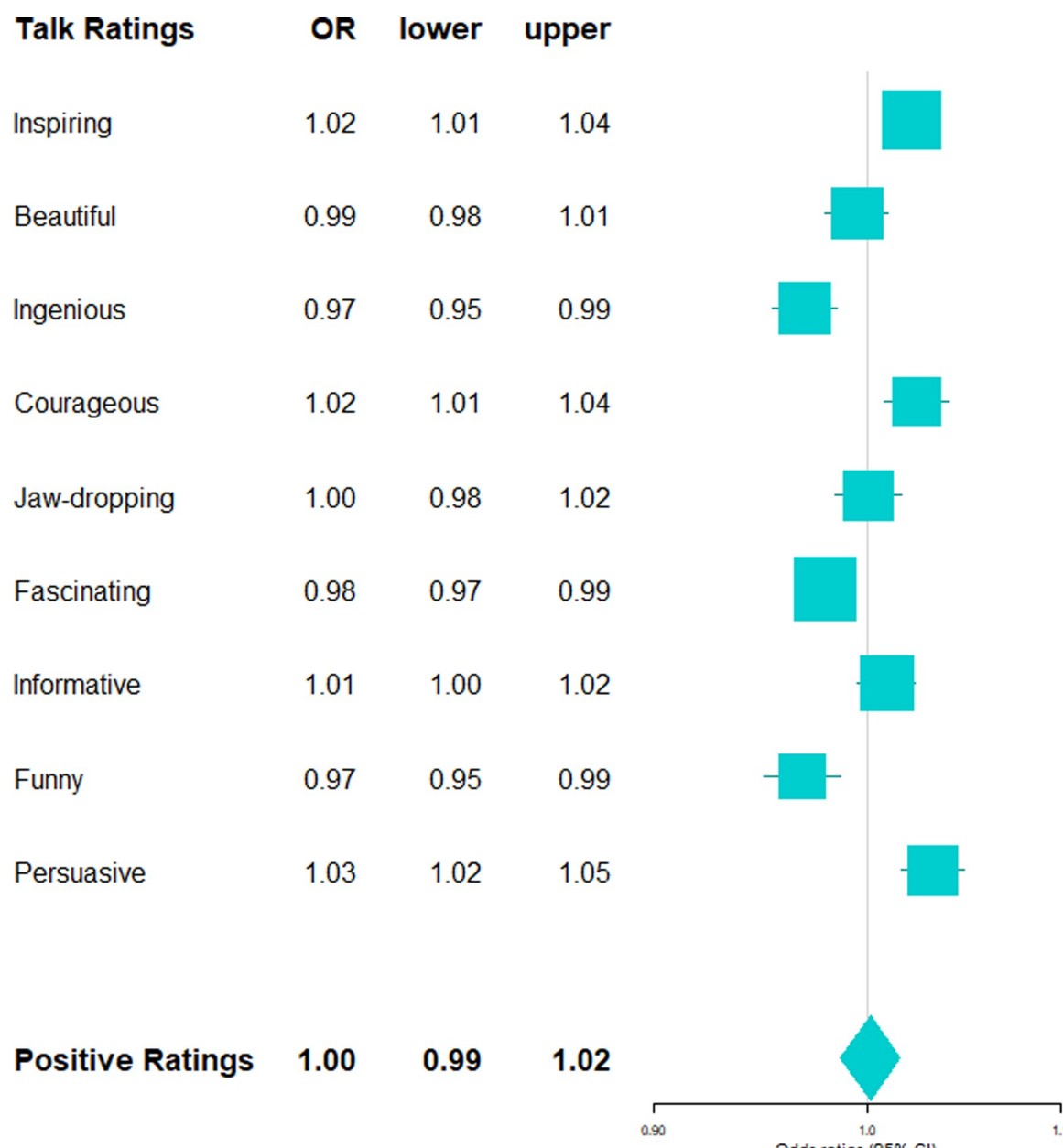

**Fig 6. Positive talk ratings by age-linked language style.** Exponentiated regression coefficients (OR: Odds ratios) and 95% confidence intervals (lower: lower bound, upper: upper bound) from beta regressions accounting for control variables (age linear and squared, gender, academic status, time online, number of ratings). OR > 1 indicate an increased likelihood for talks with more senior language style to receive the rating type. Positive talk ratings (boldface) refers to the aggregated score of all positive ratings.

not supported. The lack of interaction effects between gender and gendered language (see Tables G and H in S1 File for details) moreover suggests that the effects of gendered language did not differ between male and female speakers. Congruity of TED Speakers' language style with their own gender did not seem to play an important role for talk impact, and the *congruent is prudent-hypothesis* was therefore not supported either. In fact, the results suggest that over and beyond speaker's gender, female-typical language links to greater talk impact in

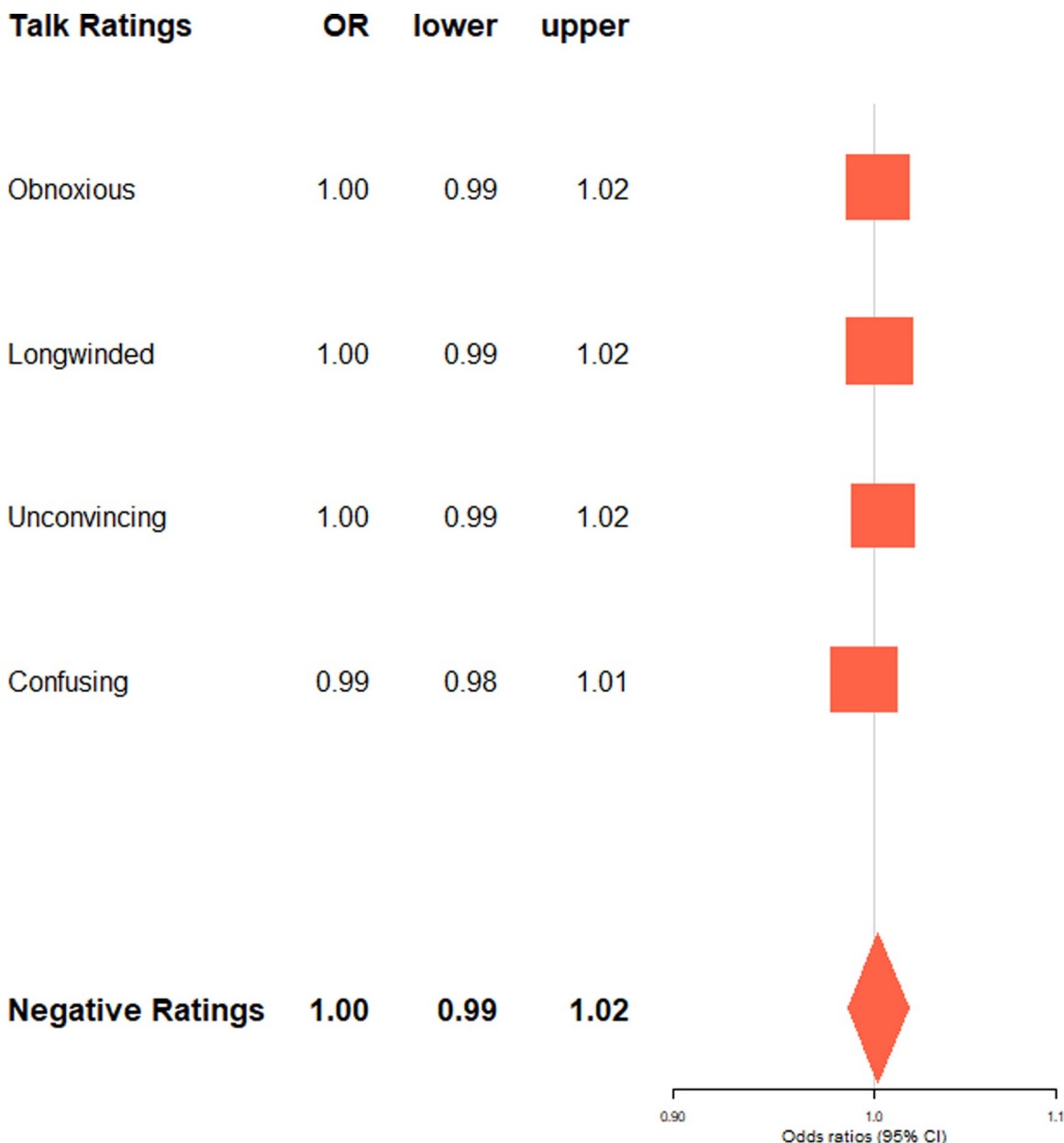

**Fig 7. Negative talk ratings by age-linked language style.** Exponentiated regression coefficients (OR: Odds ratios) and 95% confidence intervals (lower: lower bound, upper: upper bound) from beta regressions accounting for control variables (age linear and squared, gender, academic status, time online, number of ratings). OR > 1 indicate an increased likelihood for talks with more senior language style to receive the rating type. Negative talk ratings (boldface) refers to the aggregated score of all negative ratings.

terms of views, and that female and male-typical language styles were both uniquely linked to different facets of positive talk ratings.

## Question 2. Do TED Speakers' chronological age and age-linked language style predict talk impact?

**Age, age-linked language and impact quantity.**   Neither speaker's chronological age nor age-linked language style were associated with the number of talk views and thus to the overall

quantity of talk impact. We report the detailed results of age on impact quantity in Table E in S1 File.

**Age-linked language and impact quality.** Regarding impact quality, the main results of age-linked language style are depicted in Figs 6 and 7, that is the odds ratios (with 95% confidence intervals) for age-linked language in predicting positive and negative talk ratings. Additional details on the main results for this research question are reported in Tables K and L in S1 File.

Our first assumption that older speaker's talks receive in general fewer positive and more negative ratings found little support, as chronological age was associated with neither aggregated scores of positivity ($p$ = .847), nor negativity ($p$ = .345) of talk ratings (see Tables K and L in S1 File for details). Older speakers' talks were, however, more likely to be rated as "long-winded" ($B$ = 0.01, 95% CI = 0.001; 0.01; $p$ = .030). There were also significant quadratic effects of age ("beautiful": $B$ = 0.0004, 95% CI = 0.0001; 0.001; $p$ = .022, "courageous"": $B$ = 0.0003, 95% CI = 0.0000; 0.001; $p$ = .043, "longwinded" $B$ = -0.0004, 95% CI = -0.001; -0.0000; $p$ = .032). For example, talks given by middle-aged speakers' received the most "informative" ($B$ = -0.001, 95% CI = -0.001; -0.0002; $p$ = .004) ratings. This may likely reflect beliefs that middle-aged speakers tend to be at the peak of their career and competence.

Regarding the effects of age-linked language style on impact quality, more senior language style was associated with neither the aggregated scores of positivity ($p$ = .855), nor negativity ($p$ = .790) of talk ratings (see Figs 6 and 7). When looking at the different qualities of ratings separately, senior language style was not linked to any of the negative ratings (Fig 7). However, it was linked to different facets of positive ratings, such as more"inspiring"($B$ = 0.02, 95% CI = 0.01; 0.03; $p$ = .002),"courageous"($B$ = 0.02, 95% CI = 0.01; 0.04; $p$ = .004) and"persuasive"($B$ = 0.03, 95% CI = 0.02; 0.05; $p$ < .001) ratings, but fewer"ingenious"($B$ = -0.03, 95% CI = -0.05; -0.02; $p$ < .001),"fascinating"($B$ = -0.02, 95% CI = -0.03; -0.01; $p$ = .001), and"funny"($B$ = -0.03, 95% CI = -0.05; -0.01; $p$ = .001) ratings (Fig 6).

Average marginal effects (AME) for speaker's age and age-linked language style are reported in Tables M and N in S1 File. As an example, TED Talks, on average, received 19.35%"inspiring"ratings (Table 2). The AME of age-linked language style on"inspiring"was 0.003 (see Table M in S1 File). A one unit increase in age-linked language style (corresponding to a one year increase in age estimated from language) was thus linked to an 0.3% increase in"inspiring"ratings, if the other model predictors were held constant.

In similar ways as gender-linked language, age-linked language style therefore uniquely predicted facets of positive talk impact, and these effects were above and beyond speaker's chronological age.

## Additional analysis

Following the suggestion of a reviewer, we conducted additional analyses on speakers' gender- and age-linked language style and impact quantity (i.e., talk views) of their TED Talks. More specifically, because women typically have a more dynamic or narrative (as opposed to analytical) language style [44,46], the goal of this additional analysis was to get a more fine-grained picture of what aspects of female language style drove the success of TED Talks. For this reason, we added an index of dynamic-analytical language [44] as a control variable to our analyses. Female language style and analytical language correlated at r = -.28 in our sample. The full model results including all control variables are presented in Table O in S1 File.

The quantile regression results showed that analytical language was negatively associated with the number of talk views in all quantiles ($p$ < .010) except the 10% quantile ($p$ = .069). This suggests that TED Talks were watched more often, the more dynamic (rather than

analytic) their language style was. Again, the slopes were steeper among the highest quantiles (see Table O in S1 File), thus suggesting that this relationship between dynamic language and talk views was particularly evident among the most popular talks. When controlling for dynamic-analytical language, however, female language failed to reach statistical significance. This may be taken to suggest that it was mainly the dynamic and narrative character of female-typical language style that predicted TED Talk views.

We moreover repeated the analysis for age-linked language style and talk views while controlling for analytical language. The full model results for this additional analysis are presented in Table P in S1 File. Senior language and analytical language correlated at r = .25 in our sample. Similar to the previous model, analytical language style showed negative associations with talk views in all quantiles ($p < .050$). Age-linked language style was not related to the number of talk views in any of the quantiles. In line with the main analyses reported, this corroborates the finding that speakers' younger versus senior language styles are not related with the impact quantity of their TED Talks.

## Discussion

Social processes may fundamentally bias human social perception, and women and older people are typically disadvantaged when it comes to social evaluations. In this study, we examined how such processes may guide social evaluations and impact in the digital age–contexts in which social influence is largely potentiated compared to offline settings. We considered language as a unique manifestation of gender- and age-prototypical behavior through which social evaluations and impact are shaped. In other words, we investigated implicit processes of how age- and gender-linked behaviors trigger evaluations and whether these evalations are in line with expectations from theoretical frameworks on gender and age stereotypes. This study used TED as a large and particularly successful example of modern digital communication.

Female TED Speakers were underrepresented (31.8%), and, as expected, presenting as a women was associated with lower TED Talk impact in terms of quantity (i.e. number of views). However, the use of *female language*, in fact, was associated with *higher* quantitative impact. This is in sharp contrast to our expectation *(male over female-hypothesis)* that the male language benefit commonly observed in offline interactions and written language [6,10,11,42] would generalize onto digital communication spaces. Most importantly, female-typical language was associated with more talk views irrespective of speaker's gender – Female-typical language thus conferred an advantage for male and female speakers alike in our sample. In other words, behavior typically shown by women was associated with higher talk impact. This finding contradicts the common notion that female-typical behavior elicits perceptions of warmth and that this comes at the cost of ascribed status and power. These results provided an important foundation for new explorations of the associations between female-typical behavior and status. In the social context of digital communication aimed at spreading relevant ideas, it seems to be an advantage to "speak like a woman"– perhaps because it fosters stronger connection with the audience. Further research is needed to investigate whether this advantageous effect is limited to TED Talks, digital contexts of communication, or whether rules may have changed more generally so that warmth may no longer be in contrast to power and competence when it comes to gender stereotypes. More generally, our findings open the door for speculations about how the rules underlying social influence in digital communication might have shifted from the rules in traditional forms of communication.

Moreover, the picture revealed from associations between gender- and age-linked language, and impact *quality* (rating types) seemed in line with common gender and age stereotypes on the warmth and competence dimension space. With few exceptions–i.e. "funny", "unconvincing"–

the main effects of speaker's gender and chronological age were in the same direction as those of their language markers. The effects of gender and age thus reassembled in the additional effect of gender- and age-linked language use. Associations between gender-linked language accentuations and talk ratings did not systematically differ between men and women–for impact *quality* of their talks, it did therefore not matter whether speakers' language was in line with their identified gender or not. Our *congruent is prudent hypothesis* did thus not find support in the data. In the following, we discuss the results and their implications in more detail.

## Impact quantity: Which language style links to more views?

Even though presenting as a male speaker was associated with more views, "speaking like a man" was not. Talks given in a more female-typical language style were viewed more often, and this pattern was particularly pronounced among the most influential TED talks, for which more female-typical language came with steep increases in views. The lack of relationship between female language and views in the lowest quantile suggests that for talks that were not viewed often, factors other than language might have played a more important role for their impact. This finding is in line with research suggesting that female behaviors are rewarded if they are accompanied by external cues of authority and status [21]. Female-typical language may thus be a powerful tool to promote impact and visibility particularly for speakers whose competence or status is out of question–independent of whether they are male or female.

Interestingly, the effects of speaker's gender and gender-linked language style on views were in the opposite direction, corroborating the idea that female-typical language uniquely associates with talk influence above and beyond speaker's gender. Largely replicating previous research suggesting that men are more influential than women online [1,7], this gendered pattern was again particularly evident among the most popular TED Talks in our study. This finding can be taken as a further example showing that gender stereotypes may even affect women of high status and expertise, such as TED Speakers [19,74,75]. In contrast, female-typical *behavior* was rewarded in TED Talks independently of the presenter's gender. This opens the door for further research of the particularities of digital communication spaces that might reward gendered behavior in a different way than in offline communication.

## Impact quality: Gender- and age-linked language use and facets of warmth versus competence

The effects of gender-linked language style on talk ratings did further not differ between male and female speakers, contrasting suggestions from the gender role congruence perspective [61]. Although we considered our two hypotheses, the *male over female-hypothesis* versus the *congruent is prudent-hypothesis*, as competing against each other, neither of them found support from the language data.

The results instead suggest that female and male language styles evoked *different facets* of positive and negative impact, which seemed to reflect common stereotype contents of warmth versus competence [4]. While both female gender and female language independently of each other predicted higher percentages of ratings resembling warmth, (e.g. "beautiful", "courageous"), some facets showed opposing effects for female gender and female-typical language (e.g. "funny"), and female gender seemed to evoke somewhat hostile attitudes ("obnoxious") that were not observed for female language. While previous work has shown how gender stereotypes constrain the evaluation of humor for men and women so that humorous women are ascribed lower status than non-humorous women [25], our finding that female gender links to fewer "funny" ratings may have reflected female speakers' caution in using humor in their TED Talks. In addition, our results also point to one particular aspect of female gender, namely

female-typical language style, that may be especially suitable to elicit humorous reactions from an audience, possibly because of its personal and conversational character.

In general, our results suggest distinct dimensions of "warmth" and "competence" as underlying dimensions of positive TED Talk ratings, rather than one universal positive dimension. This was against our initial expectation, but was also reflected in the low internal consistency of the aggregated positivity score. The notion of a competence dimension that associates with male stereotypes is in line with other research: Word counts of "brilliant" and "genius" in anonymous student evaluations of professors predicted the representation of women and African Americans in the corresponding academic fields [76]. The authors interpreted this as a reflection of stereotypical beliefs about white male "brilliance" across different academic fields.

We note that the rating type "courageous", which we interpreted as a "warmth" dimension, could also have a different connotation. It might alternatively reflect a patronizing praise that is often shown towards marginalized groups, such as women in male-dominated fields [77,78]. "Courageous" might then not clearly reflect a positive rating type, but also contain some degree of ambiguity.

Compared to gender, speaker's age and the language proxy of age showed less pronounced associations with talk impact. While a more senior language style did not predict the number of talk views, it did predict facets of positive ratings. These differences seemed in line with age stereotypes that often simultaneously contain both positive and negative contents, such as wisdom and warmth [4,14,29,79]: "Inspiring"; "courageous", and "persuasive" were reactions more often given to talks presented in a more senior language.

Although the results of speakers' chronological age on talk ratings may have reflected somewhat negative age stereotypes ("longwinded"), the results largely suggest that–despite the broad age range (12 to 94 years) in our sample–older speakers' talks did not perform worse in terms of impact than younger speakers' talks. Older TED Speakers represent a unique sample of older adults who have been highly successful and influential throughout their careers, which may perhaps protect them from common age stereotypes. In general, gender seemed to have a more profound effect on talk impact than age.

## Female language in digital communication: Higher relevance of emotional connection?

Taken together, the results demonstrate how language represents one pathway through which stereotypes and prototypical behaviors may shape social evaluations and influence. The pronounced effects of gender-linked language style corroborate previous findings that people have internalized schemes of how men and women speak, which may in turn affect their evaluations [e.g., 55]. Female language style is not the style that has typically been used by leaders and high status individuals [51], and its link to greater talk impact was unexpected. While previous research in social media found analytical language to be more influential [42,43], our study observed benefits of more female and thus more narrative and personal language style. Female language's more personal and narrative style might convey authenticity and psychological closeness [47,53]. Accordingly, the benefit of female-typical language in the current study was driven by a more narrative style as shown in our supplemental analyses. In novel digital communication spaces in which speakers need to connect with large and diverse audiences, a narrative style may be especially relevant and female language may eventually show to advantage. In light of the at-scale influence of digital platforms, the finding that female-typical language may boost influence is of high relevance.

It is important to note, that female academics in our sample tended to have a more male-typical language style than non-academics, thus hinting at how socialization may shape gendered

language patterns. Women in academia might have implicitly adapted to more male-typical behavior, possibly to counterbalance potential negative effects of female gender. Our findings indicate that such an adaptation towards male-typical behavior might not be necessary after all, as female language may have its own benefits–at least in digital communication spaces.

While the present study focused on verbal aspects of female communication, oral presentations come with a variety of other observable behaviors. Paralinguistic cues, such as pitch, or volume may affect persuasion [80], and future research will have to determine how stylistic features of female language in interaction with paralinguistic cues might predict impact. Future research will also be required to exactly understand in which contexts (e.g.; oral versus written, academic versus non-academic) and for whom (e.g.; junior versus senior professionals) female-typical language holds the persuasive potential we observed in TED Talks.

Technological shifts pose new challenges on how to convey complex ideas to a large number of people. In the digital age, speakers must appeal to large, diverse crowds and communicate their ideas effectively while still sustaining the audience's attention and compete with alternatives. In similar ways as social interactions have been described as more socially binding processes among women [81], female-typical language may perhaps be a successful tool to increase emotional rapport with the audience, and drive the message's overall impact.

Our findings on the intriguing benefits of female-typical language might be taken as another example of a more general phenomenon recently described in how influential figures communicate [82]. Both in political and in news media contexts, researchers observed a cultural shift for leaders towards using less analytical, and more informal language. In other words, influential figures increasingly use simple rhetoric, and the effectiveness of this rhetoric in convincing a mass audience has been exemplified by the American Presidency [82]. Results from our study provide first evidence that such a shift might extend onto digital contexts in which influence may be achieved at an unprecedented scale, and that this may be in favor of female language. Women might then be the winners of this cultural shift by having a language style that fits with the rhetoric requirements of the digital culture.

The present research adds to the extensive body of literature on social perception in two meaningful ways. First, we showed that language commonly linked to social groups may uniquely link to social evaluation and influence, and might thus explain one mechanism through which social evaluations occur. Second, we showed that social processes, e.g. gendered evaluations, often observed in offline settings might partially extend onto modern, digital contexts, but in other ways than expected. Digital communication contexts might be more receptive of female behavior, and represent spaces in which femininity may unfold its full potential.

## Limitations and outlook

The research presented in this article should be understood in the context of its limitations. First, although conveying ideas in a concise and entertaining manner is probably relevant for most public speeches, TED Talks may have a particularly pronounced entertaining character and the degree to which findings from this study generalize onto other public speech and modern self-presentation contexts will be subject to further research. Furthermore, TED coaches its speakers on various aspects of presentation techniques [83]. Although TED Speakers' language use shows gender differences in line with those observed in more spontaneous contexts [38], their language might not fully conform with their natural language use. This should, however, not change conclusions drawn from the present study, since we investigated the effects of gender-linked language styles independently of speakers' genders.

And third, we investigated the effects of gender- and age-linked language styles separately from each other to create a preliminary understanding these language signatures' unique

effects on talk impact. In real life, these language styles do of course not occur in isolation of each other, but speakers do rather have different combinations of prototypical female, male, and senior language styles that possibly even depend on context. While the correlation between these two language metrics was small ($r$ = .13), studying their interaction was out of scope but might be an interesting avenue for future research.

Based on the present work, the following future directions can be taken into consideration. The current study took a naturalistic approach to examine how gender- and age-linked language styles link to social evaluations. A promising future research line on this topic will be studies that employ experimental manipulations. Future studies could for example present participants with the same talk given in the same language style by a male versus a female speaker to participants in order to gain a more fine-grained picture on the interplay between gender, language style and social evaluations. Experimental work could also vary the quality of the talks in order to shed light on how subjective talk ratings (e.g., "informative") correspond with objective measures, or whether they mainly represent biases.

Regarding the gender congruence hypothesis, the present study investigated congruence in terms of speaker's gender and gender-linked language styles. However, there are also other types of congruence worth looking at in this area. One fruitful future avenue could be to examine how congruence between gender-linked topic and gender-linked language style relates to social evaluations. Previous research showed that audiences may prefer language that violates their expectations, such that songs with lyrics that were atypical for their genre were more popular [84]. Linking this to gender-role expectations, future studies could investigate whether male language links to more positive ratings in talks about early childhood education (a stereotypically "female" topic) as compared to talks about technology (a stereotypically "male" topic) – or the other way around. Regarding the main effects of speaker's gender, another study showed that gendered TED audience evaluations did not alter when taking different talk topics into account [85]. This suggests that at least with respect to speaker's gender, gender–topic congruence did not matter as the finding that female speakers receive less positive evaluations generalized across different talk topics [85].

Finally, we note that while influential theories on gender identity describe "masculinity" and "femininity" as independent from each other to some degree (e.g., a person can score high on typical "masculine" as well as "feminine" traits [26,27]), language-based measures of gender are often based on unidimensional conceptualizations of gender (i.e., continuous scores ranging from "masculine" to "feminine"). Future research should also explore gender-linked language styles while taking the multi-dimensional structure of language into account. While this was out of scope of the present article, it would be especially promising to conduct longitudinal studies to examine how an individual's female-typical or male-typical language styles varies from one situation to another. This would also allow to infer how associations between language style and social evaluations generalize from one interaction context to another. Empirically speaking, however, the unidimensional approach has been shown to fit language data well as demonstrated by high predictive validity in terms of the often dispayed binary gender identity [12].

## Conclusion

The present study provided promising first insights into how online influence is shaped by speakers' language use, and how language styles more typically used by women may drive talk impact in novel digital settings. A female language style may thus represent a powerful tool that men and women alike could take advantage of to generate views and influence others online. This findings are especially intriguing because they might suggest that modern

communication contexts provide new spaces in society in which typical female behavior is rewarded and may go hand in hand with the perception of high status, which has not always been the case in traditional professional domains [61].

While our findings might be a symbol of a general phenomenon in which leaders increasingly use more informal language, future research will have to determine how the results of the present study generalize onto other contexts and populations. Nonetheless, the study yields first insights into how digital contexts might hold a bright future in which female language is heard. Since gendered language styles are thought to be most pronounced in spontaneous rather than in constrained speech contexts [46], and may be adapted to in certain situations [38], a promising future avenue will be to see whether speakers can explicitly be trained to manipulate their own language style in order to boost visibility of their message. Future research will have to evaluate conditions and possible boundary effects for the advantage of female language style in online speech contexts.

## Supporting information

**S1 File. Impact quantity and quality: Detailed overview of descriptive statistics, results of main analyses and supplemental analyses.**
(DOCX)

## Acknowledgments

The authors would like to acknowledge Vanessa Infanger for her valuable support in data preparation, and Robert G. Moulder for his insightful advice on some of the data analytical procedures. Moreover, the authors would like to thank two anonymous reviewers for their helpful comments.

During her work on this project, Tabea Meier was a pre-doctoral fellow of LIFE (International Max Planck Research School on the Life Course; participating institutions: MPI for Human Development, Humboldt-Universität zu Berlin, Freie Universität Berlin, University of Michigan, University of Virginia, University of Zurich).

Tabea Meier and Andrea B. Horn are affiliated with, and Mike Martin is the director of the URPP"Dynamics of Healthy Aging"at the University of Zurich, Switzerland.

## Author Contributions

**Conceptualization:** Tabea Meier, Ryan L. Boyd, Matthias R. Mehl, Andrea B. Horn.

**Data curation:** Tabea Meier.

**Formal analysis:** Tabea Meier.

**Investigation:** Tabea Meier.

**Methodology:** Tabea Meier, Andrea B. Horn.

**Resources:** James W. Pennebaker, Mike Martin, Andrea B. Horn.

**Software:** Ryan L. Boyd.

**Supervision:** Ryan L. Boyd, Matthias R. Mehl, Anne Milek, James W. Pennebaker, Mike Martin, Andrea B. Horn.

**Validation:** Ryan L. Boyd, Matthias R. Mehl, Andrea B. Horn.

**Visualization:** Tabea Meier.

Writing – **original draft:** Tabea Meier.

Writing – **review & editing:** Tabea Meier, Ryan L. Boyd, Matthias R. Mehl, Anne Milek,
James W. Pennebaker, Mike Martin, Markus Wolf, Andrea B. Horn.

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
