## [Decision Letter · Decision Letter 0]

7 Oct 2020

PONE-D-20-08183

Stereotyping in the digital age: Male is „ingenious“, female is „beautiful"

PLOS ONE

Dear Dr. Meier,

Thank you for submitting your manuscript to PLOS ONE. After careful consideration, we feel that it has merit but does not fully meet PLOS ONE’s publication criteria as it currently stands. Therefore, we invite you to submit a revised version of the manuscript that addresses the points raised during the review process.

We look forward to receiving your revised manuscript.

Kind regards,

Marte Otten, Ph.D.

Academic Editor

PLOS ONE

Journal Requirements:

2. In your Methods section, please include additional information about your dataset and ensure that you have included a statement specifying whether the collection method complied with the terms and conditions for the websites from which you have collected data.

3. We noted in your submission details that a portion of your manuscript may have been presented or published elsewhere.

[Part of the data (i.e. transcripts of TED Talks and a translated subsample) had been used in another manuscript (see uploaded copy) recently published in "Social Psychological and Personality Science". The manuscript presents a separate research question and the analyses are distinct. It can therefore be excluded that this is a dual publication.]

Additional Editor Comments (if provided):

Both reviewers are enthousiastic about your manuscript (perhaps they would rate it "informative" or even "persuasive" or "fascinating" ;-). I would like to highlight a few of the issues raised by reviewer 1 and 2.

Reviewer 1 makes an important point about whether the way you approach the data-analysis really allows you to conclude anything about stereotypes (male vs female) or whether this approach is more model-based, and less about (cultural) stereotypes and assumptions surrounding gender. I am looking forward to reading your reply to this, as I find this to be a relevant consideration.

I also would love to see, together with reviewer 1, a slightly more in depth discussion about the bigger picture is probably a good idea. I would assume this is a topic that interests a broad audience, and the way you interpret these findings and their meaning for how men/women are seen online could be rather influential - I think taking that step and putting those ideas out there with this paper is worth it!

Reviewer 2, in addition to a number of interesting theoretical approaches which are likely to enrich your introduction and discussion, suggests two additional/ alternative analyses for your data; taking a new perspective on what gender incongruence might mean in your dataset (perhaps more related to topic than to the gender of the speaker) and taking into account that the use of analytic/dynamic langauge might also play a role/ be a confounding factor in your dataset : I am looking forward to seeing the outcome of these analyses.

Reviewers' comments:

Reviewer's Responses to Questions

**Comments to the Author**

1. Is the manuscript technically sound, and do the data support the conclusions?

Reviewer #1: Yes

Reviewer #2: Yes

2. Has the statistical analysis been performed appropriately and rigorously? 

Reviewer #1: Yes

Reviewer #2: Yes

3. Have the authors made all data underlying the findings in their manuscript fully available?

Reviewer #1: Yes

Reviewer #2: Yes

4. Is the manuscript presented in an intelligible fashion and written in standard English?

Reviewer #1: Yes

Reviewer #2: Yes

5. Review Comments to the Author

Reviewer #1: This is an very well conceived of and well written paper. I thought it was clear in its aims and transparent in its approach. The methodology applied is rigorous and the use of data very intelligent. I think this would make a fine contribution to the field. However, I do have some comments which I think should be addressed before publication.

Line 254 – you say you collected the videos and meta data in March 2018, but what were the dates that the videos themselves were given / uploaded? I think this is important context

Line 265 – I personally think it looks a little odd to give two decimal places to ‘views’

Line 295 – Can you expand upon why you chose the 1st of January to measure the age from here?

Section around line 310 – Here you seem to suggest that the data dictionary you apply here is based on past studies which reliably predicted the characteristics of the speaker based on their language. That is, this data dictionary was not necessarily based on established stereotypes? If this is the case ( and if I have mis understood please disregard), could not the title of this paper be considered slightly inaccurate as it seemingly suggests you will be measuring language stereotypes rather than just established gendered language difference? I think these are different concepts so at the very least some expansion of this should be included.

Line 331 – I think this is a potentially problematic definition of ‘academic’. Although rarer in recent years, there are still many academics without doctorates

Lines 573 onward – If I’m reading this correctly, unpacking this does your analysis mean that if, at the aggregate level women garner less ‘impact’ by your measures but ‘female language’ garners more impact, does this mean it is men who deploy female language who confer the biggest advantage? Is there a way to clearly split this out to test directly or at least add some controls? It would be fascinating. I think this would really help enhance your already very good analysis.

In terms of the conclusions, I think there needs to be a slight expansion of the bigger picture here. What does it mean more broadly that women speakers receive less positive impact for their talks but female language seemingly confers an advantage?

Reviewer #2: I think this is a fascinating paper. The topic is important, and I appreciate the wealth of naturalistic data on both sides (speaker language and audience ratings). My only comments have to do with potential alternate interpretations of some of the findings and additional (in my mind optional) analyses to support or rule out those interpretations.

1) I was surprised to see that the gender congruence hypothesis (research question 1b) didn’t show any gendered language x speaker gender effects. It is difficult to interpret a null result, but I expected audiences to like feminine speech more in the context of a male or masculine speaker. TED audiences presumably go to the site for new and exciting information (“ideas worth spreading”). Thus they may appreciate it when a speaker’s language violates their expectations. That would at least be consistent with processing fluency research suggesting that in some artistic or intellectual areas (e.g., visual art or rap lyrics), audiences may prefer more challenging, disfluent stimuli to information that’s easy to process (e.g., Belke et al., 2015; Berger et al., 2018).

Rather than looking at gender congruence (do men talk like men?), it might be more fruitful to look at congruence with the talk's area (do people from masculine fields talk like men?). Feminine language may be viewed especially positively when it violates expectations for the topic, regardless of the speaker’s gender. For example, neuroscientist Jill Bolte Taylor’s TED talk, “My Stroke of Insight,” may be as popular as it is partly because, contrary to expectations about neuroscience, it is very dynamic and conversational, focusing quite a bit on emotions, relationships, and abstract connections among all people (to be clear, I have not actually LIWCed that talk).

References:

Belke, B., Leder, H., & Carbon, C. C. (2015). When challenging art gets liked: Evidences for a dual preference formation process for fluent and non-fluent portraits. PloS one, 10(8), e0131796.

Berger, J., & Packard, G. (2018). Are atypical things more popular?. Psychological science, 29(7), 1178-1184.

2) Audiences' positive views of feminine speech reminded me of Jordan and Pennebaker’s analytic-dynamic index and the cultural shift towards more dynamic thinking in the US, as reflected in the language of US presidents (citation [78] in the manuscript). As the authors note, women tend to use more dynamic language (more conversational, less formal; more verbs than nouns) and thus could be considered "the winners of this cultural shift” (pp. 30-31). Can you control for analytic-dynamic language without removing the essential components of female-typical language? For example, could you control for verbs or test verb rates as a possible moderator or order to disentangle feminine from dynamic speech?

3) To me, “courageous” seems less relevant to the “warmth” dimension of the stereotype content model and more like the kind of patronizing praise that emphasizes a person’s outgroup. TED talks typically focus on male-dominated fields, such as science, business, and activism. Even in social science or environmentalism, which are more stereotypically feminine or gender-inclusive, men often have more power or dramatically outnumber women at the highest ranks (e.g., full professors in psychology). Saying that a female scientist is courageous (but not ingenious) could be seen as suggesting that women must be brave (but perhaps not very wise) to persist in a field where they’re on the margins or atypical. It reminds me of the backlash to female-specific praise that’s sometimes seen in the media or real life (e.g., Serena Williams not wanting to be praised as a groundbreaking female athlete, preferring instead to be judged on her merits as an athlete irrespective of gender or race). More empirically, what I’m talking about is similar to research on leaders in male-typical environments giving women high praise but fewer resources (which not surprisingly elicits anger from women):

Gervais, S. J., & Vescio, T. K. (2012). The effect of patronizing behavior and control on men and women’s performance in stereotypically masculine domains. Sex Roles, 66(7-8), 479-491.

Vescio, T. K., Gervais, S. J., Snyder, M., & Hoover, A. (2005). Power and the creation of patronizing environments: the stereotype-based behaviors of the powerful and their effects on female performance in masculine domains. Journal of personality and social psychology, 88(4), 658.

Is it possible to disentangle whether “courageous” ratings reflect warmth, patronizing praise, stereotypes about the degree to which a person belongs in or is typical for their field, or simply the topics of the TED talks (that is, maybe women are just talking about adversity they’ve faced more often than men)? For example, do you see similar ratings for authors from other marginalized groups, like people of color or people of atypical ages (speakers in their teens / 20s or 80s+)?

Independent of exploring the “courageous” results further, it could be useful to code the talks for topic and topic gender associations (whether the topic is viewed as stereotypically masculine or feminine). You might either control for those variables or consider topic as another way in which gendered language can be congruent or incongruent.

4) I would be interested in hearing more about how you interpret the results for the “funny” ratings (where being male and using more feminine language are both associated with being rated as funnier). I am not a humor expert, but I think that humor research shows that funny things tend to be absurd, surprising, or unexpected. Is that what’s happening here – that people think men who use feminine language are funny partly because their speaking style is surprising? Or is the feminine language-“funny” rating correlation not moderated by speaker gender?

5) In the Method section, you note that, “We consider the gender score as a measure for gender style prototypicality in language. Negative values on the score refer to a more male-typical language style, and positive values to a more female-typical language style.” (p. 14)

I think that measure is fine – it’s well-validated, and it makes sense considering that most demographic scales in psychology treat gender as binary (male or female). However, gender is arguably multidimensional, with masculinity and femininity negatively correlated but not orthogonal (e.g., some of Janet Spence’s research). To anticipate reviewers and readers who will see that unidimensional measure as short-sighted, it would be nice to see a section in the Introduction discussing the multidimensionality of gender and the challenges in translating those dimensions to linguistic measures that are based on binary gender labels.

References:

Eagly, A. H., & Wood, W. (2017). Janet Taylor Spence: Innovator in the study of gender. Sex Roles, 77(11-12), 725-733.

Spence, J. T. (1993). Gender-related traits and gender ideology: evidence for a multifactorial theory. Journal of personality and social psychology, 64(4), 624.

6) You note that “TED coaches its speakers” (p. 31). What does this coaching involve? I didn’t see a reference for this statement. Is the coaching tailored? Do women and men potentially get different advice, depending on who prepares them for their talk? Are all speakers asked to play up their personal stories, emphasizing any adversity they have overcome (perhaps magnifying gender differences in perceptions of courage or bravery)?

7) This is more of a future direction than an idea that should be shoehorned into this paper, but it would be nice to be able to show that gender differences in “informative” ratings are factually untrue. Is there a simple way to automatically measure informativeness or information density in these transcripts (for example, through entropy or lexical diversity)?

8) Minor notes:

On page 11, I believe that “gender-conform language use” should be “gender conforming” or “gender congruent.”

There are some irregular spacing and alignment issues that should be corrected before publishing (e.g., alternating single and double spaces between paragraphs, centered paragraphs on pages 30-31).

The abstract is relatively long -- it might be more effective if it were condensed.

Some of the writing was a little informal and colloquial (“short end of the stick,” p. 26). I don’t mind occasional informality in empirical papers, but colloquialisms might be challenging for non-native English speakers.

6. PLOS authors have the option to publish the peer review history of their article (what does this mean?). If published, this will include your full peer review and any attached files.

Reviewer #1: No

Reviewer #2: No

---

## [Author Response · Author response to Decision Letter 0]

16 Nov 2020

Additional Editor Comments (if provided):

Both reviewers are enthousiastic about your manuscript (perhaps they would rate it "informative" or even "persuasive" or "fascinating" ;-). I would like to highlight a few of the issues raised by reviewer 1 and 2.

***

Thank you very much for the helpful comments. We highly appreciate your and the reviewers’ enthusiasm about the manuscript and are grateful for the suggestions that we believe helped to make the manuscript stronger.

***

Reviewer 1 makes an important point about whether the way you approach the data-analysis really allows you to conclude anything about stereotypes (male vs female) or whether this approach is more model-based, and less about (cultural) stereotypes and assumptions surrounding gender. I am looking forward to reading your reply to this, as I find this to be a relevant consideration.

***

Thank you for the opportunity to clarify. As we elaborate below in the response to the reviewer’s comment, we used dictionaries to quantify speakers’ age- and gender-linked language styles, i.e. the degree to which they have a typical female/male or young/senior language style. These dictionaries developed by Sap et al. (2014) have previously been used to estimate a given speaker’s age and gender from their language and are based on earlier work underlining robust gender differences in language use. We then use these language scores in our models to predict the impact of the TED Talks (talk ratings and views). In other words, we use language measures that could be interpreted as manifestations of age- and gender-typical behavior, and investigate the associations they have with the possibly stereotype-informed, social evaluative outcomes of the talks. We do not directly investigate explicit (gender or age) stereotypic beliefs or assumptions. We do, however, relate the social evaluative outcomes with dimensions of gender and age stereotyping established in the literature (warmth, competence). 

In order to clarify, we explicitly introduce this in the introduction section (line 64):

“Findings like these support the assumption that language patterns are one behavioral feature that makes social groups such as gender or age salient and trigger stereotyping. Gender- and age-linked language have been quantified by deriving general language patterns that empirically link to the social groups of gender and age [12,13].”

We furthermore changed the wording in the Discussion (line 617): 

“We considered language as a unique manifestation of gender- and age-prototypical behavior through which social evaluations and impact are shaped. In other words, we investigated implicit processes of how age- and gender-linked behaviors trigger evaluations and whether these evalations are in line with expectations from theoretical frameworks on gender and age stereotypes.” and in the Abstract (line 28): “Our goal was to investigate how gender- and age-linked language styles – beyond chronological age and identified gender – link to talk impact and whether this reflects gender and age stereotypes.” 

We additionally elaborated a sentence in the corresponding Methods section (line 330): “We consider the gender and age scores as measures for gender- and age-style prototypicality in language “and use them to predict talk ratings and views in order to infer the role of gender- and age-linked language in stereotypical social evaluations.” 

Following the reviewer’s suggestion, we moreover slightly altered the title of the manuscript: Stereotyping in the digital age: Male language is “ingenious”, female language is “beautiful” – and popular

***

I also would love to see, together with reviewer 1, a slightly more in depth discussion about the bigger picture is probably a good idea. I would assume this is a topic that interests a broad audience, and the way you interpret these findings and their meaning for how men/women are seen online could be rather influential - I think taking that step and putting those ideas out there with this paper is worth it!

***

Thank you for pointing this out. In order to make the conclusion clearer reagarding what it means that speaker’s gender and gender-linked language style show opposing effects regarding talk impact, we elaborated on this in the discussion (line 628): “Most importantly, female-typical language was associated with more talk views irrespective of speaker’s gender � Female-typical language thus conferred an advantage for male and female speakers alike in our sample. In other words, behavior typically shown by women was associated with higher talk impact. This finding contradicts the common notion that female-typical behavior elicits perceptions of warmth and that this comes at the cost of ascribed status and power. These results provide an important foundation for new explorations of the associations between female-typical behavior and status. In the social context of digital communication aimed at spreading relevant ideas, it seems to be an advantage to “speak like a woman” � perhaps because it fosters stronger connection with the audience. Further research is needed to investigate whether this advantageous effect is limited to TED Talks, digital contexts of communication, or whether rules may have changed more generally so that warmth may no longer be in contrast to power and competence when it comes to gender stereotypes.”

Furthermore, regarding the reviewer’s questions on how to interpret the results from our analysis, we provide a more detailed, direct response to this item below in the context of Reviewer 1’s last comment. 

***

Reviewer 2, in addition to a number of interesting theoretical approaches which are likely to enrich your introduction and discussion, suggests two additional/ alternative analyses for your data; taking a new perspective on what gender incongruence might mean in your dataset (perhaps more related to topic than to the gender of the speaker) and taking into account that the use of analytic/dynamic langauge might also play a role/ be a confounding factor in your dataset : I am looking forward to seeing the outcome of these analyses.

***

Thank you for this suggestion. We reran our analyses controlling for the “analytical” language index from the text analysis program LIWC. We provide a detailed response on this in Reviewer 2’s second comment. 

1) Analytical language

The results of the additional analyses showed that “analytical language” had consistent negative associations with talk impact (number of views), suggesting that indeed a more dynamic / conversational language style was linked to more talk views. Female-typical language style did not have a significant association above and beyond (i.e. controlling for) analytical language. Importantly, this does not alter our conclusions but rather provides additional clues as to the linguistic mechanisms underlying female-typical language and further empirical support to our conclusions: As stated in the introduction (lines 178-179), women tend to have a more dynamic/conversational (= less analytical) language style, and the additional analyses suggest that it is this dynamic/conversational aspect of female-typical language that drives talk success on TED. 

We report the results of the additional analyses at the end of the Results section under “additional analyses” (and in Tables O and P in S1 File). We have also uploaded the data and analysis scripts for this additional analysis to an OSF repository (https://osf.io/qkm6u/) to allow readers to independently explore this aspect.

We furthermore discussed these results by adding the following sentence to the Discussion section (line 729): “Accordingly, the benefit of female-typical language in the current study was driven by a more narrative style as shown in our supplemental analyses.”

2) Gender congruence with topics

Regarding the reviewer’s second suggestion of an alternative perspective on gender congruence: We agree that this would be highly interesting. However, unfortunately, we are unable to conduct this additional analysis with the data we have at hand. We believe that conducting this additional analysis in a way that is methodologically sound and allows the anticipated inferences would go well beyond the scope of the manuscript, and we are happy to provide this as a suggestion for future work.

As we elaborate below in the reviewer’s comment, our dataset contains information on the topics of TED Talks in form of tags. There are, however, about 400 different tags (e.g., “health”, “brain”; “culture”) and it would not be straightforward to categorize them by gender with high accuracy and reliability (e.g., is “culture” male or female?). This is especially so because most talks contain multiple tags that may even “contradict” each other in their gender prototypicality (e.g., a talk can be labelled as both “technology” and “children”). 

Therefore, we see this as an interesting future avenue for this research but are unfortunately not able to provide a methodologically sound answer to this question within the scope of the present manuscript.

In our revised manuscript, we outline this idea as a future direction in the Discussion section (line 800): “Regarding the gender congruence hypothesis, the present study investigated congruence in terms of speaker’s gender and gender-linked language styles. However, there are also other types of congruence worth looking at in this area. One fruitful future avenue could be to examine how congruence between gender-linked topic and gender-linked language style relates to social evaluations. Previous research showed that audiences may prefer language that violates their expectations, such that songs with lyrics that were atypical for their genre were more popular [84]. Linking this to gender-role expectations, future studies could investigate whether male language links to more positive ratings in talks about early childhood education (a stereotypically “female” topic) as compared to talks about technology (a stereotypically “male” topic) - or the other way around. Regarding the main effects of speaker’s gender, another study showed that gendered TED audience evaluations did not alter when taking different talk topics into account [85]. This suggests that at least with respect to speaker’s gender, gender–topic congruence did not matter as the finding that female speakers receive less positive evaluations generalized across different talk topics [85].”

***

Reviewer #1: This is an very well conceived of and well written paper. I thought it was clear in its aims and transparent in its approach. The methodology applied is rigorous and the use of data very intelligent. I think this would make a fine contribution to the field. However, I do have some comments which I think should be addressed before publication.

***

We would like to thank the reviewer for the interesting and helpful comments and suggestions. We value the reviewers’ appreciation of our manuscript’s relevance to the field, and it is clear that working through this reviewer’s feedback hasfurther improved our manuscript.

***

Line 254 – you say you collected the videos and meta data in March 2018, but what were the dates that the videos themselves were given / uploaded? I think this is important context

***

Thank you for pointing this out. The talks in our sample had been delivered between 1990 and 2017 (the vast majority between 2001 and 2017), and thus covered almost the whole timespan of TED, which was founded in 1984. TED started to post videos of talks online in 2006. In all analyses, we controlled for how long the talks had been online prior to data collection. We added a sentence in our Method section to give readers more context information (line 276): “Overall, our final sample included TED Talks that had been delivered between 1990 and 2017. “ 

***

Line 265 – I personally think it looks a little odd to give two decimal places to ‘views’

***

Thank you – We totally agree and removed the decimals.

***

Line 295 – Can you expand upon why you chose the 1st of January to measure the age from here?

***

Thank you for this question. We agree that this may have been not very clearly described. We calculated “speaker’s age” as the difference between their “date of birth” and “date the talk was given”. As stated in the manuscript, our primary goal was to obtain speakers exact birthdate to obtain precise measures of their age. The exact birthdate could, however, not be found for all speakers, but oftentimes only the year they were born was available from internet searches. Since the talks were given at different times of the year, we used January 1 in case only the year they were born was known. This arbitrary date was used to ensure that there were no systematic biases, meaning that for all speakers for whom only the year (and not the date) of birth was known, simply the year of birth was taken as reference. We acknowledge that another way would have been to use mid-year (e.g., July 1) instead of January 1 for speakers for whom the exact birthdate was unknown. Most importantly, however, since we used the same date for everyone, this did not provoke any systematic biases. We added a sentence in the manuscript and hope that this helps to clarify (line 315): “In other words, if only the year, but not the exact date of birth was known, we took the year of birth as reference to infer their age at the time the talk was given.”

***

Section around line 310 – Here you seem to suggest that the data dictionary you apply here is based on past studies which reliably predicted the characteristics of the speaker based on their language. That is, this data dictionary was not necessarily based on established stereotypes? If this is the case ( and if I have mis understood please disregard), could not the title of this paper be considered slightly inaccurate as it seemingly suggests you will be measuring language stereotypes rather than just established gendered language difference? I think these are different concepts so at the very least some expansion of this should be included.

***

Thank you for the opportunity to clarify. Indeed, the dictionaries we applied have in past studies been used to reliably estimate speakers’ gender and age from their language. In our study, we use the dictionaries as a measure for the gender/age “prototypicality” of their language style: that is, the degree to which they have a female/male-typical or senior/younger language style. We then use these language scores to predict talk impact (talk ratings and views). In other words, we are not measuring explicit stereotypes of a target individual but, rather, prototypical female/male and younger/senior language, respectively — these behavioral manifestations (e.g., female-typical language) are then linked to the social evaluative outcomes of the talks.

We agree, that the former title might have been misleading in regard and have now updated it to: Stereotyping in the digital age: Male language is “ingenious”, female language is “beautiful” – and popular

As outlined above, we clarified this by consistently referring to gender- and age-linked (or female-typical) language (rather than “stereotypical langauge”) and introduce a definition in the introduction (line 64):

“Findings like these support the assumption that language patterns are one behavioral feature that makes social groups such as gender or age salient and trigger stereotyping. Gender- and age-linked language have been quantified by deriving general language patterns that empirically link to the social groups of gender and age [12,13].”

We moreover elaborated a sentence in the corresponding Methods section (line 330):“ We consider the gender and age scores as measures for gender- and age-style prototypicality in language and use them to predict talk ratings and views in order to infer the role of gender- and age-linked language in stereotypical social evaluations.” 

***

Line 331 – I think this is a potentially problematic definition of ‘academic’. Although rarer in recent years, there are still many academics without doctorates

***

We fully agree about the potentially problematic aspects of this definition of “academic”. The reason we relied upon this definition is that it had been previously used by another study that also examined TED Talks (Sugimoto et al., 2013). This is stated in the manuscript (line 348:“ This procedure was informed by previous findings showing that public trust in scientists is high for researchers in academia [67], and that TED presenters’ academic status links to their talk impact [7]. … Following the procedure proposed by [7], …”). 

Since the study by Sugimoto et al. reported associations between speaker’s academic status and other measures of talk success similar to those in our study (i.e., comments on the videos), we wanted to make sure to control for “academic status” in our analyses by using a measure that conforms to the study by Sugimoto et al. We now added a sentence to make this more specific (line 356: “This allowed us to control for speakers’ academic status in a way that conforms to prior research in the context of TED Talks (see [7]).”

Reference:

Sugimoto CR, Thelwall M, Larivière V, Tsou A, Mongeon P, Macaluso B (2013) Scientists Popularizing Science: Characteristics and Impact of TED Talk Presenters. PLoS ONE 8(4): e62403. https://doi.org/10.1371/journal.pone.0062403

***

Lines 573 onward – If I’m reading this correctly, unpacking this does your analysis mean that if, at the aggregate level women garner less ‘impact’ by your measures but ‘female language’ garners more impact, does this mean it is men who deploy female language who confer the biggest advantage? Is there a way to clearly split this out to test directly or at least add some controls? It would be fascinating. I think this would really help enhance your already very good analysis.

In terms of the conclusions, I think there needs to be a slight expansion of the bigger picture here. What does it mean more broadly that women speakers receive less positive impact for their talks but female language seemingly confers an advantage?

***

This is a great question, and we agree that this is a central point to clarify within the manuscript. In our analysis, we have directly tested for the possibility that (for example) male speakers with female-typical language confer the biggest advantage. Our results suggest that this is not the case, as the interaction effect “speaker’s gender * female language style” did not reach statistical significance (see Table C in S1 File). Rather, our results suggest that, irrespective of whether the speaker is male or female, speakers with female-typical language garner more impact (i.e., talk views). 

We elaborated more on this question in the discussion section in order to further clarify the conclusion (line 628): “Most importantly, female-typical language was associated with more talk views irrespective of speaker’s gender – Female-typical language thus conferred an advantage for male and female speakers alike in our sample. In other words, behavior typically shown by women was associated with higher talk impact. This finding contradicts the common notion that female-typical behavior elicits perceptions of warmth and that this comes at the cost of ascribed status and power. These results provide an important foundation for new explorations of the associations between female-typical behavior and status. In the social context of digital communication aimed at spreading relevant ideas, it seems to be an advantage to “speak like a woman” perhaps because it fosters stronger connection with the audience. Further research is needed to investigate whether this advantageous effect is limited to TED Talks, digital contexts of communication, or whether rules may have changed more generally so that warmth may no longer be in contrast to power and competence when it comes to gender stereotypes.”

***

Reviewer #2: I think this is a fascinating paper. The topic is important, and I appreciate the wealth of naturalistic data on both sides (speaker language and audience ratings). My only comments have to do with potential alternate interpretations of some of the findings and additional (in my mind optional) analyses to support or rule out those interpretations.

***

We would like to thank the reviewer for the thoughtful, constructive comments and suggestions. We value the reviewers’ appreciation of our manuscript and tried to address the suggestions, which we think helped to further enrich our manuscript.

***

1) I was surprised to see that the gender congruence hypothesis (research question 1b) didn’t show any gendered language x speaker gender effects. It is difficult to interpret a null result, but I expected audiences to like feminine speech more in the context of a male or masculine speaker. TED audiences presumably go to the site for new and exciting information (“ideas worth spreading”). Thus they may appreciate it when a speaker’s language violates their expectations. That would at least be consistent with processing fluency research suggesting that in some artistic or intellectual areas (e.g., visual art or rap lyrics), audiences may prefer more challenging, disfluent stimuli to information that’s easy to process (e.g., Belke et al., 2015; Berger et al., 2018).

Rather than looking at gender congruence (do men talk like men?), it might be more fruitful to look at congruence with the talk's area (do people from masculine fields talk like men?). Feminine language may be viewed especially positively when it violates expectations for the topic, regardless of the speaker’s gender. For example, neuroscientist Jill Bolte Taylor’s TED talk, “My Stroke of Insight,” may be as popular as it is partly because, contrary to expectations about neuroscience, it is very dynamic and conversational, focusing quite a bit on emotions, relationships, and abstract connections among all people (to be clear, I have not actually LIWCed that talk).

References:

Belke, B., Leder, H., & Carbon, C. C. (2015). When challenging art gets liked: Evidences for a dual preference formation process for fluent and non-fluent portraits. PloS one, 10(8), e0131796.

Berger, J., & Packard, G. (2018). Are atypical things more popular?. Psychological science, 29(7), 1178-1184.

***

Thank you for this highly interesting suggestion. Unfortunately, with the dataset we have, it is not feasible for us to conduct an analysis that takes a gendered language * gendered topic interaction into account in an empirically rigorous manner. We do have information on the topics of the TED Talks: The talks have tags, but there are about 400 different tags (e.g., “health”, “brain”; “culture”) and it would not be straightforward to categorize them for gender with high accuracy (e.g., is “culture” a male or female topic?). This is especially so because most talks contain multiple tags that may even “contradict” each other in terms of gender prototypicality (e.g., several talks include both the labels of “war” [stereotypically male] and “parenting” [stereotypically female]). Therefore, we see this as a very interesting future avenue for this research but are not able to provide a meaningful answer to this question within the scope of the present study. 

Nevertheless, we agree that this is a rather interesting and meaningful question and, as such, we mention this future direction in the Discussion section (line 800): “Regarding the gender congruence hypothesis, the present study investigated congruence in terms of speaker’s gender and gender-linked language styles. However, there are also other types of congruence worth looking at in this area. One fruitful future avenue could be to examine how congruence between gender-linked topic and gender-linked language style relates to social evaluations. Previous research showed that audiences may prefer language that violates their expectations, such that songs with lyrics that were atypical for their genre were more popular [84]. Linking this to gender-role expectations, future studies could investigate whether male language links to more positive ratings in talks about early childhood education (a stereotypically “female” topic) as compared to talks about technology (a stereotypically “male” topic) – or the other way around. Regarding the main effects of speaker’s gender, another study showed that gendered TED audience evaluations did not alter when taking different talk topics into account [85]. This suggests that at least with respect to speaker’s gender, gender–topic congruence did not matter as the finding that female speakers receive less positive evaluations generalized across different talk topics [85].”

***

2) Audiences' positive views of feminine speech reminded me of Jordan and Pennebaker’s analytic-dynamic index and the cultural shift towards more dynamic thinking in the US, as reflected in the language of US presidents (citation [78] in the manuscript). As the authors note, women tend to use more dynamic language (more conversational, less formal; more verbs than nouns) and thus could be considered "the winners of this cultural shift” (pp. 30-31). Can you control for analytic-dynamic language without removing the essential components of female-typical language? For example, could you control for verbs or test verb rates as a possible moderator or order to disentangle feminine from dynamic speech?

***

Thank you for this suggestion. Ultimately, the degree to which it is possible to partial out the variance associated with the Analytic—Dynamic index without removing the essential components of female-typical language is not clear. Based on past work, we do know that more dynamic language use is indeed associated with being female. Indeed, in our sample, female-typical language and the Analytic—Dynamic language index (on which higher scores indicate a more analytical language style) correlated at r = -.283. This association shows, however, that the broader constellation of psychosocial components that constitute female-typical language goes beyond being analytic or dynamic in one’s language. 

To hone in a bit more precisely on this question, we reran our analyses controlling for the “analytical” language index from the text analysis program LIWC. What this analysis tells us, then, is to what degree analytic/dynamic language serves as an important component of female-typical language that also meaningfully impacts the social cognition of audience members. The results of the additional analyses showed that “analytical language” had a consistent, negative association with talk impact (number of views), suggesting that indeed a more dynamic / conversational language style was linked to more talk views. Female-typical language style did not have a significant association above and beyond (i.e. controlling for) analytical language. Our analyses thus suggest that a dynamic language styles does share variance with female-typical language style on talk popularity. Importantly, we note that this does not alter our conclusions but, rather, provides some preliminary clues into the mechanisms that may constitute the effect of female-typical language and further empirical support for our conclusions: As stated in the introduction, women tend to have a more dynamic/conversational language style and the additional analyses suggest that it is this dynamic/conversational aspect of female-typical language that drives talk success on TED. 

We report the results of the additional analyses at the end of the Results section (and in Tables O and P in S1 File). Like the material for the other analyses, we uploaded the data/analysis script for this additional analysis to OSF (https://osf.io/qkm6u/). 

We furthermore discussed this finding adding the following passage to the Discussion section (line 729): “Accordingly, the benefit of female-typical language in the current study was driven by a more narrative style as shown in our supplemental analyses.”

***

3) To me, “courageous” seems less relevant to the “warmth” dimension of the stereotype content model and more like the kind of patronizing praise that emphasizes a person’s outgroup. TED talks typically focus on male-dominated fields, such as science, business, and activism. Even in social science or environmentalism, which are more stereotypically feminine or gender-inclusive, men often have more power or dramatically outnumber women at the highest ranks (e.g., full professors in psychology). Saying that a female scientist is courageous (but not ingenious) could be seen as suggesting that women must be brave (but perhaps not very wise) to persist in a field where they’re on the margins or atypical. It reminds me of the backlash to female-specific praise that’s sometimes seen in the media or real life (e.g., Serena Williams not wanting to be praised as a groundbreaking female athlete, preferring instead to be judged on her merits as an athlete irrespective of gender or race). More empirically, what I’m talking about is similar to research on leaders in male-typical environments giving women high praise but fewer resources (which not surprisingly elicits anger from women):

Gervais, S. J., & Vescio, T. K. (2012). The effect of patronizing behavior and control on men and women’s performance in stereotypically masculine domains. Sex Roles, 66(7-8), 479-491.

Vescio, T. K., Gervais, S. J., Snyder, M., & Hoover, A. (2005). Power and the creation of patronizing environments: the stereotype-based behaviors of the powerful and their effects on female performance in masculine domains. Journal of personality and social psychology, 88(4), 658.

Is it possible to disentangle whether “courageous” ratings reflect warmth, patronizing praise, stereotypes about the degree to which a person belongs in or is typical for their field, or simply the topics of the TED talks (that is, maybe women are just talking about adversity they’ve faced more often than men)? For example, do you see similar ratings for authors from other marginalized groups, like people of color or people of atypical ages (speakers in their teens / 20s or 80s+)?

***

This is a meaningful question, and we were also curious to explore this further. Although we do not know what users actually meant when they rated talks as “courageous” (or how that meaning varied by individual users), we note that “courageous” showed positive, small-to-moderately sized correlations (statistically significant) with other positive rating types reflecting “warmth”: “inspiring” (r = .38) and “beautiful” (r = .08; see intercorrelations of ratings in Table B in S1 File). Moreover, “courageous” had negative correlations with all rating types we had categorized as “negative”. 

At the same time, “courageous” ratings also show negative correlations with some positive rating types that reflect “competence”, such as e.g. “fascinating” (r = -.51), “informative” (r = -.25), suggesting that talks that were rated as “courageous” indeed also received fewer ratings reflecting competence. 

Altogether, this strengthens the assumption that, in this dataset or context, “courageous” represents a “positive” rating reflecting “warmth”, although there is likely some degree of nuance/variability in how, precisely, the viewers applied this label.

We added this interesting discussion point to the discussion section (line 701): “We note that the rating type “courageous”, which we interpreted as a “warmth” dimension, could also have a different connotation. It might alternatively reflect a patronizing praise that is often shown towards marginalized groups, such as women in male-dominated fields [77,78]. “Courageous” might then not clearly reflect a positive rating type, but also contain some degree of ambiguity.”

***

Independent of exploring the “courageous” results further, it could be useful to code the talks for topic and topic gender associations (whether the topic is viewed as stereotypically masculine or feminine). You might either control for those variables or consider topic as another way in which gendered language can be congruent or incongruent.

***

Thank you for this suggestion. Please refer to our response to the reviewer’s comment 1.)

***

4) I would be interested in hearing more about how you interpret the results for the “funny” ratings (where being male and using more feminine language are both associated with being rated as funnier). I am not a humor expert, but I think that humor research shows that funny things tend to be absurd, surprising, or unexpected. Is that what’s happening here – that people think men who use feminine language are funny partly because their speaking style is surprising? Or is the feminine language-“funny” rating correlation not moderated by speaker gender?

***

This is an interesting question, albeit one with considerable nuance and complexity. It is indeed the case that research often characterizes humor as something of a “pleasant surprise,” however, the degree to which subtle variations in language style constitute a surprise in the classical sense (i.e., a sudden, unexpected, and pleasant transpiring that defies one’s predictions or worldview) is difficult to say, and seems to be somewhat unlikely.

Empirically, analyses of our data found that the female language/“funny” association was not moderated by speaker’s gender in our analyses, as indicated by the non-significant speakers’ gender*female language interaction effect (p = .716; Table G in S1 File). This suggests that, when speaker’s gender and gender-linked language style were controlled for one another, talks given in female-typical language style was linked to more “funny” ratings. A possible explanation for this is the more personal and dynamic/conversational character of female-typical language style that may be more suitable for eliciting humor than the more fact-based and analytical male language style. 

At the same time, presenting as a female speaker (as opposed to a male speaker) was associated with fewer “funny” ratings. This suggests that, beyond female-typical language style, other characteristics of female speakers are less likely to elicit humor or that female speakers were less likely to use humor in their talks. In general, this fits with studies showing how gender and gender stereotypes constrain the use of humor, in which humorous women were ascribed lower status than non-humorous men, whereas humorous men were ascribed higher status than non-humorous men [25]. Women or female professionals might thus be more cautious in using humor in order to avoid potential negative evaluations. 

We elaborated on this in the discussion section (line 686) to offer a more elaborated interpretation: “While previous work has shown how gender stereotypes constrain the evaluation of humor for men and women so that humorous women are ascribed lower status than non-humorous women [25], our finding that female gender links to fewer “funny” ratings may have reflected female speakers’ caution in using humor in their TED Talks. In addition, our results also point to one particular aspect of female gender, namely female-typical language style, that may be especially suitable to elicit humorous reactions from an audience, possibly because of its personal and conversational character.”

Reference:

Evans, J. B., Slaughter, J. E., Ellis, A. P. J., & Rivin, J. M. (2019). Gender and the evaluation of humor at work. Journal of Applied Psychology, 104(8), 1077-1087. http://dx.doi.org/10.1037/ apl0000395

***

5) In the Method section, you note that, “We consider the gender score as a measure for gender style prototypicality in language. Negative values on the score refer to a more male-typical language style, and positive values to a more female-typical language style.” (p. 14)

I think that measure is fine – it’s well-validated, and it makes sense considering that most demographic scales in psychology treat gender as binary (male or female). However, gender is arguably multidimensional, with masculinity and femininity negatively correlated but not orthogonal (e.g., some of Janet Spence’s research). To anticipate reviewers and readers who will see that unidimensional measure as short-sighted, it would be nice to see a section in the Introduction discussing the multidimensionality of gender and the challenges in translating those dimensions to linguistic measures that are based on binary gender labels.

References:

Eagly, A. H., & Wood, W. (2017). Janet Taylor Spence: Innovator in the study of gender. Sex Roles, 77(11-12), 725-733.

Spence, J. T. (1993). Gender-related traits and gender ideology: evidence for a multifactorial theory. Journal of personality and social psychology, 64(4), 624.

***

Thank you for raising this important and very interesting point. We added this information in the the introduction and in a section in the discussion section to illuminate this limitation of measures of gender-linked language styles:

Line 108: “It is important to note that most studies in this field, however, rely on a traditional unidimensional perspective on gender, in which femininity and masculinity are mutually exclusive, a framework that has been questioned in the past [26,27].”

Line 813:

“Finally, we note that while influential theories on gender identity describe “masculinity” and “femininity” as independent from each other to some degree (e.g., a person can score high on typical “masculine” as well as “feminine” traits [27,26]), language-based measures of gender are often based on unidimensional conceptualizations of gender (i.e., continuous scores ranging from “masculine” to “feminine”). Future research should also explore gender-linked language styles while taking the multi-dimensional structure of language into account. While this was out of scope of the present article, it would be especially promising to conduct longitudinal studies to examine how an individual’s female-typical or male-typical language styles varies from one situation to another. This would also allow to infer how associations between language style and social evaluations generalize from one interaction context to another. Empirically speaking, however, the unidimensional approach has been shown to fit language data well as demonstrated by high predictive validity in terms of the often dispayed binary gender identity [12].”

***

6) You note that “TED coaches its speakers” (p. 31). What does this coaching involve? I didn’t see a reference for this statement. Is the coaching tailored? Do women and men potentially get different advice, depending on who prepares them for their talk? Are all speakers asked to play up their personal stories, emphasizing any adversity they have overcome (perhaps magnifying gender differences in perceptions of courage or bravery)?

***

According to documentation on TED’s website, there appear to be several “best practices” that are recommended and come up in the coaching process, including things like avoiding moving too much (nervous pacing) or too little (awkward stillness), overstating points, anecdotal evidence, and so on:

https://www.ted.com/participate/organize-a-local-tedx-event/tedx-organizer-guide/speakers-program/prepare-your-speaker/rehearsals

This has also been summarized in a book by Jeremey Donovan (referred to as [83] in our manuscript.

While it is the case that TED provides coaching to its speakers, we cannot say definitively what all this coaching might involve, the universality of the coaching, or whether coaching styles/advice vary as a function of individual coaches or speaker characteristics such as age, gender, field of expertise, and so on. Additionally, we cannot speak to the extent to which TED’s coaching/guidance has changed over time or may vary across venues.

We elaborated on the sentence in question and added a citation (line 778):

“Furthermore, TED coaches its speakers on various aspects of presentation techniques [83].

***

7) This is more of a future direction than an idea that should be shoehorned into this paper, but it would be nice to be able to show that gender differences in “informative” ratings are factually untrue. Is there a simple way to automatically measure informativeness or information density in these transcripts (for example, through entropy or lexical diversity)?

***

This is a very interesting thought. We are afraid that we are unable to show whether more “informative” ratings reflect an actual higher level of informativeness or rather a bias in terms of the raters perception. We speculate that it is probably a mix of both. What we can say, however, is that “informative” ratings in our sample showed at least small correlations with other language measures we would expect to relate with informativeness. 

More specifically, talks rated as “informative” tended to show a slightly higher overall word count (r = .073; p = .015). Moreover, “informative” had a negative correlation (r = -.24, p <.001) with the “dictionary words” of the text analysis program LIWC (Pennebaker et al., 2015). Since the LIWC dictionaries mostly contain frequently used words, this may be taken to suggest that talks rated as “informative” tended to include more uncommon words, and therefore to possibly also contain more new/diverse information. 

We included the reviewer’s thought in the “limitations/outlook” section (line 792): “The current study took a naturalistic approach to examine how gender- and age-linked language styles link to social evaluations. A promising future research line on this topic will be studies that employ experimental manipulations. Future studies could for example present participants with the same talk given in the same language style by a male versus a female speaker to participants in order to gain a more fine-grained picture on the interplay between gender, language style and social evaluations. Experimental work could also vary the quality of the talks in order to shed light on how subjective talk ratings “e.g., “informative”) correspond with objective measures, or whether they mainly represent biases.”

***

8) Minor notes:

On page 11, I believe that “gender-conform language use” should be “gender conforming” or “gender congruent.”

***

Thank you for pointing this out—we changed it to “gender congruent” in the manuscript.

***

There are some irregular spacing and alignment issues that should be corrected before publishing (e.g., alternating single and double spaces between paragraphs, centered paragraphs on pages 30-31).

***

Thank you for this feedback — we re-edited spacing and alignment as suggested.

***

The abstract is relatively long -- it might be more effective if it were condensed.

***

Thank you for the suggestion. We edited the abstract in order to check for clarity and conciseness and shortened it a bit.

***

Some of the writing was a little informal and colloquial (“short end of the stick,” p. 26). I don’t mind occasional informality in empirical papers, but colloquialisms might be challenging for non-native English speakers.

***

Thank you, we agree that colloquial expressions may provoke a challenge to non-native English readers. We changed the corresponding expression to “are typically disadvantaged when it comes to social evaluations" (line 615).

***

---

## [Editor Report · Decision Letter 1]

25 Nov 2020

Stereotyping in the digital age: Male language is „ingenious“, female language is „beautiful" – and popular

PONE-D-20-08183R1

Dear Dr. Meier,

We’re pleased to inform you that your manuscript has been judged scientifically suitable for publication and will be formally accepted for publication once it meets all outstanding technical requirements.

Kind regards,

Marte Otten, Ph.D.

Academic Editor

PLOS ONE
---

## [Editor Report · Acceptance letter]

1 Dec 2020

PONE-D-20-08183R1 

Stereotyping in the digital age: Male language is “ingenious”, female language is “beautiful” – and popular 

Dear Dr. Meier:

I'm pleased to inform you that your manuscript has been deemed suitable for publication in PLOS ONE. Congratulations! Your manuscript is now with our production department. 

Kind regards, 

on behalf of

Dr. Marte Otten 

Academic Editor

PLOS ONE